



# Detecting dominant changes in irregularly sampled multivariate water quality data sets

**Christian Lehr[1,2], Ralf Dannowski[1], Thomas Kalettka[1], Christoph Merz[1,3], Boris**

**Schröder[4,5], Jörg Steidl[1] and Gunnar Lischeid[1,2]**

[1]{Leibniz Centre for Agricultural Landscape Research (ZALF), Müncheberg, Germany}

[2]{University of Potsdam, Institute for Earth and Environmental Sciences, Potsdam, Germany}

[3]{Institute of Geological Sciences, Workgroup Hydrogeology, Freie Universität Berlin,

Germany}

[4]{Landscape Ecology and Environmental Systems Analysis, Institute of Geoecology, Technische Universität Braunschweig, Langer Kamp 19c, 38106 Braunschweig, Germany}

[5] {Berlin-Brandenburg Institute of Advanced Biodiversity Research (BBIB), Altensteinstraße 6, 14195 Berlin, Germany}

Correspondence to: C. Lehr (lehr@zalf.de)



## Abstract

Time series of catchment water quality often exhibit substantial temporal and spatial variability which can rarely be traced back to single causal factors. Numerous anthropogenic and natural drivers influence groundwater and stream water quality, especially in regions with high land use intensity. In addition, typical existing monitoring data sets, e.g. from environmental agencies, are usually characterized by relatively low sampling frequency and irregular sampling in space and / or time. This complicates the differentiation between anthropogenic influence and natural variability as well as the detection of changes in water quality which indicate changes of single drivers. Detecting such changes is of fundamental interest for water management purposes as well as for scientific analyses.

We suggest the new term 'dominant changes' for changes in multivariate water quality data that concern 1) more than a single variable, 2) more than one single site and 3) more than short-term fluctuations or single events and present an exploratory framework for the detection of such 'dominant changes' in multivariate water quality data sets with irregular sampling in space and time. Firstly, we used a non-linear dimension reduction technique to derive multivariate water quality components. The components provide a sparse description of the dominant spatiotemporal dynamics in the multivariate water quality data set. In addition, they can be used to derive hypotheses on the dominant drivers influencing water quality. Secondly, different sampling sites were compared with respect to median component values. Thirdly, time series of the components at single sites were analysed for seasonal patterns and linear and non-linear trends. Spatial and temporal heterogeneities are efficiently used as a source of information rather than being considered as noise. Besides, non-linearities are considered explicitly. The approach is especially recommended for the exploratory assessment of existing long term low frequency multivariate water quality monitoring data.

We tested the approach with a large data set of stream water and groundwater quality consisting of sixteen hydrochemical variables sampled with a spatially and temporally irregular sampling scheme at 29 sites in the Uckermark region in northeast Germany from 1998 to 2009. Four components were derived and





interpreted as 1) the agriculturally induced enhancement of the natural background level of solute concentration, 2) the redox sequence from reducing conditions in deep
groundwater to post oxic conditions in shallow groundwater and oxic conditions in stream water, 3) the mixing ratio of deep and shallow groundwater to the streamflow and 4) sporadic events of slurry application in the agricultural practice. Dominant changes were observed for the first two components. The changing intensity of the $1^{st}$ component during the course of the observation period was interpreted as
response to the temporal variability of the thickness of the unsaturated zone. A steady increase of the $2^{nd}$ component throughout the monitoring period at most stream water sites pointed towards progressing depletion of the denitrification capacity of the deep aquifer.



## 1 Introduction

Numerous high frequency sampling studies unravelled the high temporal variability of stream water quality (e.g., Kirchner et al., 2004; Cassidy and Jordan, 2011; Halliday et al., 2012; Neal et al., 2012; Wade et al., 2012; Aubert et al., 2013; Kirchner and Neal, 2013; Tunaley et al. 2016; Rode et al., 2016; Blaen et al., 2017). Therefore, monitoring water quantity and quality on the timescale of the hydrological
response of the catchment is a key requirement for understanding water quality dynamics and its driving processes in detail (Kirchner et al., 2004; Neal et al., 2012; Halliday et al., 2012). While the development of sensor technology, data loggers and transmission technology hopefully will help to significantly increase the number of high-frequency monitoring programmes in the future, most of the existing monitoring
programmes so far applied a rather low sampling frequency. Nonetheless, there is common agreement that for short periods with high-frequency data, longer periods of low-frequency monitoring provide invaluable context (Burt et al., 2011; Neal et al., 2012; Halliday et al., 2012; Bieroza et al., 2014). This is especially true for existing long term records which are required as reference to distinguish between natural
short term and long term variability of the observed variables and the assessment of the effects of anthropogenic influence on water quality such as changes in land use in the catchment (Burt et al., 2008; Howden et al., 2011).

The intriguing temporal and spatial variability in water quality monitoring data sets can in most cases hardly be related to single causal factors. Instead, a variety of
biogeochemical processes and anthropogenic influences interact at different scales impeding identification of clear cause-effect relationships (e.g., Stumm and Morgan, 1996; Neal, 2004; Scanlon et al., 2007; Raymond et al., 2008; Basu et al., 2010; Basu et al., 2011; Aubert et al., 2013; Kroeze et al., 2013; Beudert et al., 2015). Usually a single solute is affected by numerous different drivers at different scales
(cf., e.g., Molenat et al., 2008; Lischeid et al., 2010; Schuetz et al., 2016 for $NO_3^-$). Inversely, a single driver usually has an impact on various solutes (Massmann et al., 2004; Lischeid and Bittersohl, 2008). This suggests that trend analyses of single variables might easily be misleading with respect to the identification of driving factors. For this purpose techniques which are able to account for the interaction of



multiple drivers and observed variables are preferable.

On the other hand, despite their complexity, catchments are highly constrained systems. Usually only a few processes are dominant and determining the main dynamics of stream flow, groundwater head or water quality (Grayson and Blöschl, 2000; Sivakumar, 2004; Lischeid et al., 2016). Using joint information from different
solutes is an established way to derive hypotheses on processes or other causal factors that are dominant in the monitored data. For this purpose, dimension reduction techniques, especially the linear principal component analysis (PCA), have been used in analyses of multivariate water quality data for long, mostly as exploratory tool for descriptive process identification (e.g. Usunoff and Guzmán-
Guzmán, 1989; Haag and Westrich, 2002; Cloutier et al., 2008) or for determining mixing ratios (e.g., Hooper et al., 1990; Capell et al., 2011). If the analysed data consist of time series of one or several variables observed at different sites, then the temporal features of the results of the dimension reduction can be analysed in a spatially explicit way, e.g. with respect to seasonal patterns or long term
developments at the monitored sites (Lischeid and Bittersohl, 2008; Lischeid et al., 2010).

However, many of the methods commonly used for analysing temporal developments in monitoring data sets require regularly sampled data. In practice the spatiotemporal design of sampling campaigns and monitoring networks often evolves
during the sampling period in an irregular way. In order to obtain a regularly sampled data set, additional information with a different sampling design, e.g. from pilot studies or single sampling campaigns, might not be utilized in the analysis at all. Further irregularities in the spatiotemporal structure of environmental monitoring data sets arise typically during the monitoring itself from a variety of reasons such as
failure of sensors or data loggers, measurement errors, loss of samples, periods of ice or drought, etc. Thus, in environmental monitoring practice, data sets with gaps and periods with corrupted measurements are more the rule rather than the exception.

Lischeid et al. (2010) suggested a combination of exploratory data analysis
methods to detect and analyse dominant processes and their temporal development



in multivariate water quality data sets that is capable of dealing with irregular time series. We built on that and extended it towards the detection of 'dominant changes' in time series of multivariate water quality data that are monitored at different sites, i.e. at different parts of a catchment or in different catchments within a region. In analogy to the dominant process concept (Grayson and Blöschl, 2000; Sivakumar, 2004), we use the term 'dominant changes' in a broad and descriptive sense referring to systemic changes that clearly exceed the 'usual' range of heterogeneities in the temporal, spatial or inter-variable structure of the observed water quality data. We considered changes as dominant that concerned 1) main components of the multivariate water quality data set rather than single water quality variables (multivariate components); 2) behaviour at various sites rather than at single sites (multiple sites); and 3) long-term behaviour rather than short-term fluctuations or single events (long-term patterns).

To identify the dominant changes, we combined exploratory data analysis methods for non-linear dimension reduction, spectral analysis, linear and non-linear trend estimation and monotonic trend test in one exploratory framework. The suggested approach was tested with a multivariate water quality data set that has been sampled with a spatially and temporally irregular sampling scheme in northeast Germany from 1998 to 2009. In the following, we present and discuss the results of our case study according to the three aspects of 'dominant changes': 1) multivariate components, 2) multiple sites and 3) long-term patterns. We continue with a discussion of 4) effects of the irregular sampling and 5) methodological aspects of the exploratory framework.

## 2   Data

### 2.1 Study area

The study area is the upper part of the basin of the Ucker river located in the northeast of Germany, about 90 km north of Berlin, which drains to the Baltic Sea another 50 km further north. It is part of the Leibniz Centre for Agricultural Landscape Research (ZALF) long-term monitoring region AgroScapeLab Quillow, the LTER-D (Long Term Ecological Research Network, Germany) and the TERENO (Terrestrial



Environmental Observatories, http://teodoor.icg.kfa-juelich.de) Northeastern German Lowland Observatory. Water samples have been taken in the adjacent catchments of Dauergraben (78.9 km²), Stierngraben (104.8 km²), and Quillow (399.4 km²) with its subcatchments Strom (235.8 km²) and Peege (25.6 km²) (Figure 1). For the part of

the Ucker catchment which is situated within the federal state Brandenburg a mean annual precipitation of 584.5 mm and a mean annual temperature of 8.3°C was found for the 1961-1990 period and a mean annual climatic water balance of -40.4 mm was estimated with the ARC/EGMO model (Lahmer et al., 2000). The mean climatic water balance exhibited high interannual variability with -181.4 mm in the summer half year

and +141 mm in the winter half year.

The topography of the region developed basically during the Pomerian stage and the Mecklenburgian stage of the Weichselian ice age, i.e. 15,200 to 14,100 years before present. Altitude varies from 20 m in the lowlands of the Ucker river to more than 100 m above sea level in the southwestern part of the study area. During the

Pleistocene, repeated advances and recessions of the ice sheet deposited highly heterogeneous unconsolidated sediments of about 150 m to 200 m thickness. The base consists of a thick Oligocene clay layer which separates the upper freshwater groundwater system from saline groundwater underneath. Based on borehole surveys, up to seven aquifers divided by layers of till have been identified within the

unconsolidated Quaternary sediments. In some parts of the region patches of halophilious plants are found in the lowlands indicating local upwelling of saline groundwater from the underlying Tertiary aquifer through windows of the Oligocene clay layer.

Loamy and sandy loamy soils prevail that developed from the till substrate. Most of

the region is intensively used as cropland, although the fraction of arable land differs between the catchments (Table 1). Forests comprise only a minor fraction of the area (Table 1). Land cover did not change within the study period from 1998 to 2009. The riparian zone of the catchments is mostly used as grassland, underlain by peat and organic and sandy fluvial deposits. The hummocky landscape includes about 1300

closed drainage basins and small ponds with an area of the water surface < 1 ha (Kalettka and Rudat, 2006; Lischeid et al., 2016). Many of the larger depressions



have been connected by ditches to facilitate drainage. Partly, these ditches have later been replaced by underground pipes for land reclamation. In addition, agricultural soils are extensively drained by subsurface tile drainage systems. From the 13$^{th}$

century till the end of the 19$^{th}$ century, the energy of the natural water courses was also occasionally used to power mills. Today, those mills are not active any longer and have been replaced in most cases by weirs for water management or ramps. For more details on the study site, please see Merz and Steidl (2015).





Figure 1 Map of the study area. Coordinates of UTM-zone 33N are given in m. Upper panel: Stream water monitoring sites and the location of the study area (Upper Ucker river catchment) within Germany. Lower panel: Section with the included groundwater monitoring sites. For better readability only the number of the ID of the monitoring sites is shown.




Table 1 Share of land use classes in the different catchments (percent of land cover) based on CORINE Land Cover data (2000).

| | Settlements / Industry | Arable land | Grass-land | Lakes | Others | Wet-land | Wood-land |
|---|---|---|---|---|---|---|---|
| Dauergraben | 1.7 | 92.1 | 4.1 | 1 | 0.3 | - | 0.8 |
| Ucker | 4.6 | 62.3 | 5.6 | 7.7 | 2.2 | 2.4 | 15.2 |
| Stierngraben | 1.4 | 61.2 | 15.8 | 1.2 | 0.9 | - | 19.5 |
| Strom | 2.2 | 54 | 7 | 6.9 | 1.2 | - | 28.7 |
| Quillow | 2.3 | 77 | 9.3 | 1.3 | 1.4 | - | 8.7 |
| Peege | 0 | 78.3 | 5.5 | - | - | - | 16.2 |

## 2.2 Sampling and analysis

The focus of the monitoring was the Quillow catchment. Here, eight sampling sites were located along the main stream, and another four at each of the two tributaries Peege and Strom (Figure 1 and Table S1). At the streams Dauergraben and Stierngraben and at the Ucker river, stream water quality was monitored at one site respectively. Stream water sampling started in 1998 and was performed until 2009. Groundwater quality was monitored in the Quillow catchment only, close to the middle reaches of the stream and close to the mouth of the Peege tributary, from 2000 to 2008 (Lower panel Figure 1). At this site, an up to 15 m thick horizontal till layer separates a shallow and very heterogeneous unconfined aquifer from a mainly confined deep aquifer. The separating till layer crops out further downstream (Merz and Steidl, 2015). Both aquifers were monitored (Table S2). The deep aquifer is known to be confined except at well Gd_204. Groundwater level in the deep aquifer was measured daily with automatic data loggers at wells Gd_198, Gd_201, Gd_203 and Gd_204 (Merz and Steidl, 2014a).

Groundwater quality (Merz and Steidl, 2014b) and stream water quality (Kalettka and Steidl, 2014) monitoring in the Quillow catchment covers a wide range of water quality parameters. For the multivariate analysis in this study, we considered from the joint groundwater and stream water quality data set only the 16 variables with less than 5% missing values, i.e. $NH_4^+$, $NO_3^-$, $NO_2^-$, $PO_4^{3+}$, $Na^+$, $K^+$, $Mg^{2+}$, $Ca^{2+}$, $Cl^-$, $O_2$, pH, water temperature, redox potential (Eh), electric conductivity (EC), $SO_4^{2-}$, and DOC





(Table S3). Those water samples for which more than two of the 16 monitored variables were missing were excluded from the analysis, resulting in a set of 1572 samples. In total, 0.69% of the values in the dataset were missing. In addition, we considered $HCO_3^-$ and $Fe^{2+}$ concentration from the groundwater monitoring (Table

S3).

The number of temporal replicates varied between one and 127 per site (Figure 2). In general, streams were sampled at approximately monthly intervals, and groundwater samples were taken every three months. Median [mean] sampling intervals were 29 [38.7] days for stream water and 98 [125.3] days for groundwater.

In total, sampling intervals varied between nine and 714 days (Figure 2).

Further details on the data and measurement methods are provided by Merz and Steidl (2015). The selection of water quality data used in this article and the groundwater level data have been published under CC-BY 4.0 and can be accessed at    http://open-research-data.ext.zalf.de/ResearchData/2017_340.html    and    doi:

10.4228/ZALF.2000.272 respectively.





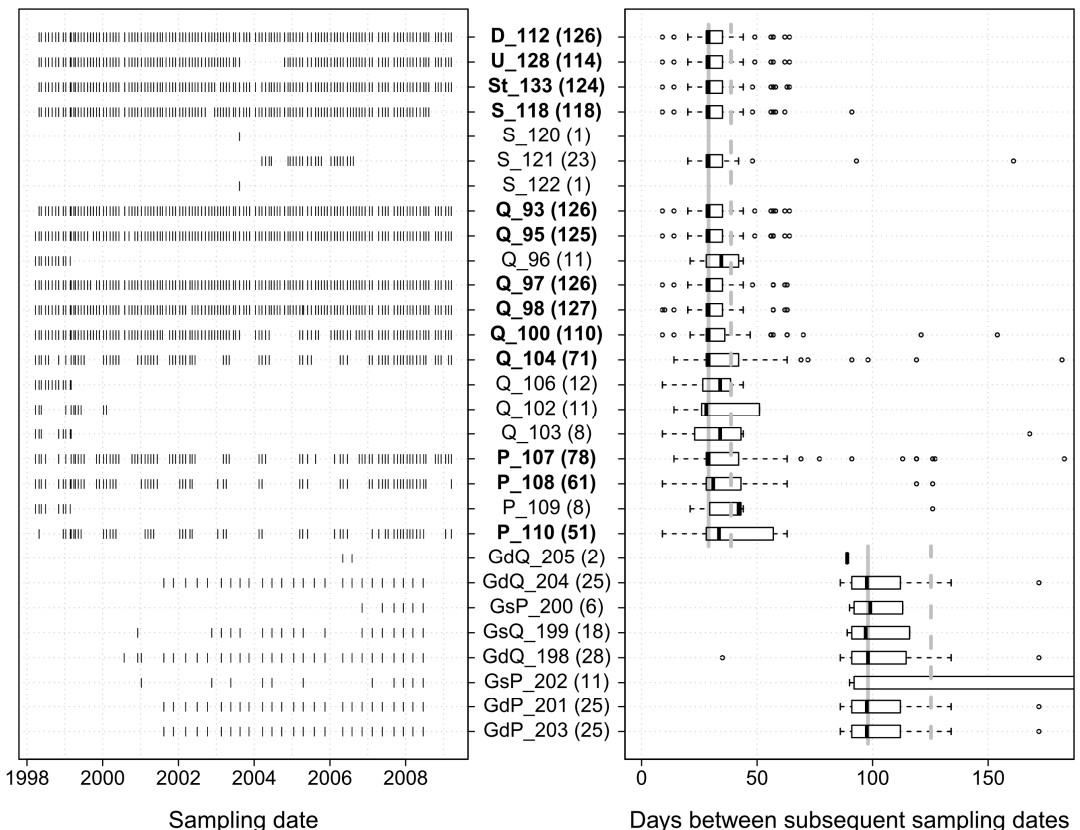

Figure 2 Left panel: Sampling dates at the sites for the whole monitoring period.
Right panel: Boxplots of the variability of sampling intervals during the monitoring
period. For better readability, the maximum of the x-axis is limited to 180 days.
Median (grey solid line) and mean (grey dashed line) of sampling intervals are shown
separately for the groundwater and stream water sites. Subscripts: P = Peege, Q =
Quillow, S = Strom, St = Stierngraben, U = Ucker, D = Dauergraben, Gs = shallow
groundwater, Gd = deep groundwater. The number of samples at each site is given in
brackets. Names of the sites with more than 50 samples are printed bold.




## 3   Methods

### 3.1 Data preprocessing

Missing values were replaced by the mean of the respective variable. This concerned at most DOC (3.44% of the values) and $NO_2^-$ (2.54%), whereas the percentage of missing values was less than 2% for each of the other 14 variables (Table S3). Values below detection limit were replaced by 0.5 times that limit. To achieve equally weighted variables the values were z-normalized to zero mean and unit standard deviation for each variable separately.

### 3.2  Exploratory framework

To identify the dominant changes, we firstly used the non-linear dimension reduction technique Isometric Feature Mapping to derive the main multivariate water quality components. To account for the interaction of groundwater and stream water, both groundwater and stream water samples have been analysed together in one joint analysis. Secondly, we studied differences between the sites with respect to median component values. Thirdly, we analysed the time series of the components at sites with more than 50 samples. Seasonal patterns were analysed with the Lomb-Scargle approach (Lomb, 1976; Scargle, 1982; Scargle, 1989) and – if significant – were subtracted from the series prior to trend analyses. Please note that the term 'seasonal' refers to the annual cycle throughout the article. Linear trends were estimated with the Theil-Sen estimator and tested for significance with the Mann-Kendall Test. Non-linear trends were depicted with the locally weighted regression (LOESS) approach (Cleveland, 1979; Cleveland and Devlin, 1988). We then related resulting low-frequency patterns to the long-term groundwater head dynamics, likewise determined as LOESS smooth of the de-seasonalised series. Time series analysis at different sites allowed to check whether long-term patterns were consistent, pointing to more general effects in the study area.

As the methods do not require regularly sampled data in space or time, we considered every sample as additional information of the spatiotemporal variability of



the observed water quality in the study area rather than noise. Consequently, irrespective of irregularities of sampling intervals at a site or differences in sampling intervals and numbers of samples between the different sites, we included as many samples in the analysis as possible to increase the informative value and support the representativeness of the study in space and time. This might lead to a bias in the

determination of the components, as well as in the estimation of the trends of the components and their significance, if deviations from a regular sampling scheme follow a systematic pattern. To check for that, we tested the distribution of sampling intervals at all sites with N > 50 (Table S1) for normality with the Shapiro-Wilk-test and the temporal development of the lengths of the sampling intervals for the whole

observation period for monotonic trends with the Mann-Kendall-test. For all tested sites a Gaussian distribution of sampling intervals as well as a monotonic trend of the length of sampling intervals during the observation period was rejected.

### 3.3 Dimension reduction

Dimension reduction methods aim to represent a data set with a given number of dimensions (here the number of measured hydrochemical variables) in a new data space with substantially less dimensions. This is achieved by projecting the data in a new ordination system which makes a more efficient use of the intrinsic structures of the data set than the original one. The axes of the new ordination system are usually

called 'components' or 'dimensions'. In the following, we will use the term 'components'. For the values of a component we will use the term 'scores'. The reduction of the dataset's dimensionality is achieved by considering only some of the new components for further analysis. The selection process is a trade-off between reduction of the dimensionality and minimizing the loss of potentially informative

structures. Typically only the first few components are selected as they depict the main structures in the data set.

In the projection, different methods focus on different aspects of the data. For example, PCA aims for maximizing variance on the first components, classical multidimensional scaling (CMDS) at preserving the interpoint distances of the input



data in the projection, and self-organizing maps (SOM) at preserving the
neighbourhood relations (topology) of the input data in the projection (Lee, 2007). In
the last years, a variety of non-linear dimension reduction methods has been
developed (Van der Maaten et al., 2009). Although being sensitive to noisy data,
Isometric Feature Mapping (Isomap; Tenenbaum et al., 2000) was one of the best

performing approaches when applied to real-world-data (Geng et al., 2005). It has
been successfully applied in environmental research disciplines, e.g. biodiversity
studies (Mahecha et al., 2007), soil sciences (Schilli et al., 2010), climatology
(Gámez et al., 2004), and biogeochemistry (Weyer et al., 2014).

### 315    3.3.1  Principal component analysis

In our study, the well-established linear principal component analysis (PCA) served
as benchmark for the non-linear Isometric Feature Mapping. PCA is one of the most
widespread dimension reduction methods going back to research of Pearson (1901)
and Hotelling (1933). For a brief introduction to PCA, please see, e.g., Jolliffe and

Cadima (2016), for a comprehensive one Joliffe (2002). PCA aims to successively
maximize the variance of the data set on the new calculated components. The scores
of the components are calculated as weighted linear combinations of the original
variables. The weights (loadings) of the linear combination define the axes of the
data space in which the data is projected. The loadings are the eigenvectors derived

from an eigenvalue decomposition of the covariance matrix of the analysed variables.
If the analysed variables are z-normalized, as was done here, their covariance matrix
is equivalent to the (Pearson) correlation matrix. The components are ordered with
decreasing size of their eigenvalues. The share of variance that is assigned to a
component is proportional to the size of its eigenvalue in relation to the sum of all

eigenvalues. Thus, the ratio of total variance that is captured by the considered
components gives a measure of performance of the PCA. PCA was performed in R
(R Core Team, 2017) with the function 'princomp' of the default package 'stats'.



### 3.3.2 Isometric Feature Mapping

Isometric feature mapping (Isomap) is a non-linear extension of CMDS. It aims to approximate the global non-linearity in a dataset by local linear fittings (Geng et al., 2005). This is done by mapping approximated geodesic interpoint-distances to an Euclidean distance matrix via a neighbourhood graph G (Tenenbaum et al., 2000). The geodesic distance between two points is the distance along the surface of a (non-linear) manifold, in contrast to the straight-line Euclidean distance (Geng et al., 2005). The neighbourhood graph G consists of segments that connect every data point to its k nearest neighbours directly via Euclidean distances. For all non-connected points the shortest path along the neighbourhood graph G is computed as the smallest sum of connected segments via the Dijkstra-algorithm (Dijkstra, 1959). This approximation of the geodesic distances allows the adaptation of G to the global non-linear structures in a data set. The only free parameter k has to be optimized by checking the performance of several runs. The more linear the data, the higher will the optimum k be. If k equals the possible number of connections of one data point to all other data points, the approximations of the geodesic distances are equal to the Euclidean distances and the Isomap results are congruent to those of CMDS and linear PCA (Gámez et al., 2004). Finally the neighbourhood graph G is embedded in the Euclidean space.

In contrast to PCA, assessing performance based on the eigenvalues of the components is not applicable for Isomap. Performance of the dimension reduction of the Isomap approach was assessed and compared to performance of the PCA by the squared Pearson correlation coefficient ($R^2$) of the interpoint distances in the high-dimensional data space and in the low-dimensional projection spanned by selected components (Lischeid and Bittersohl 2008; Lischeid et al., 2010). A perfect fit would yield a value of 1 and a value of 0 reflects no correlation between the distance matrices of the original data and of the projection. Please note, that with this measure the contribution of single components to the overall performance does not necessarily decrease monotonically with increasing order of the components, as it is the case for the eigenvalue-based performance measure of PCA. Isomap and the determination of the distance matrices were performed with the R-package 'vegan'


(Oksanen et al., 2009).

### 3.3.3 Interpretation of components

The analysis focused on those components that explained a major fraction of the
total interpoint distances. The considered components were regarded to reflect
dominant drivers influencing water quality. Here, the term 'driver' was used for
biogeochemical and hydrological processes as well as for anthropogenic influences
affecting water quality. Correspondingly we formulated a hypothesis for each
considered component. The interpretation of the components is based on analysing
(i) the correlations between measured variables and component scores as well as (ii)
spatial and temporal patterns of the scores.

Correlation between scores of a selected component $cp_x$ and values of single
variables might be blurred due to the effects of other components on the same
variable. We excluded those effects by analysing the relationships between scores of
the selected component $cp_x$ and the residuals of the multiple linear regression mlr of
the single variable $v_i$ at hand and the remaining other considered components CP\x
(residuals):

$$cor(cp_x, residuals[mlr(v_i, CP\backslash x)]) \ , \qquad\qquad (1)$$

where CP\x is the set of m considered components, without the selected
component $cp_x$, and

$$mlr(v_i, CP\backslash x) = v_i = \beta_0 \sum_{j=1}^{m} \beta_j cp_j + residuals \qquad\qquad (2)$$

To assess the relationships between components and residuals we used bivariate
scatterplots. As a measure for monotonic but not necessarily linear correlation we
used Spearman rank-correlation.




### 3.4 Time series analysis

At sites with more than 50 samples, time series of component scores were analysed for seasonal patterns, linear trends and non-linear trends. The sites were compared with respect to the identified long-term patterns to detect general patterns in the study area. The significance level for trend and frequencies in this study was set to $p \leq 0.05$. At each site, the fractions of variance of a time series that were assigned to its seasonal pattern, linear trend or non-linear trend were determined as the $R^2$ of the respective pattern with the component series. In case of significant seasonal patterns, the estimations of the trends were based on the de-seasonalised series. Accordingly, the fractions of variance assigned to the trends were determined as the $R^2$ of the trend pattern with the de-seasonalised series. The decomposition of the time series in a seasonal component and a non-linear trend derived with LOESS was inspired by the STL-approach of Cleveland et al. (1990).

### 3.4.1 Lomb-Scargle method

Standard Fourier analysis requires equidistant time series which was not given in our study. Therefore the estimation of seasonal patterns in the time series was done with the Lomb-Scargle method, which is an extension of Fourier-Analysis to the uneven-spaced case genuinely invented in astrophysics (Lomb, 1976; Scargle, 1982). The application of the Lomb-Scargle method in this study follows to a large extent the workflow suggested by Glynn et al. (2006) as well as Hocke and Kämpfer (2009). Details are given in the Appendix A. The implementation used in this manuscript can be accessed as R-script at http://open-research-data.ext.zalf.de/ResearchData/2017_340.html.

### 3.4.2 Theil-Sen estimator and Mann-Kendall test

The linear trend was estimated with the non-parametric Theil-Sen estimator which is the median of all interpoint slopes in a time series (Theil, 1950; Sen, 1968). The Mann-Kendall test (Mann, 1945; Kendall, 1990) was used to test for significant



monotonic trends. Identified trends are not necessarily linear. Being based on rank correlation, data do not have to obey any specific distribution. Please note that we did not account for the effect of overestimation of the significance of trends with the Mann-Kendall test due to autocorrelation (Yue et al., 2002). That would have required an assessment of the lag-1 autocorrelation which was hampered by the irregular

sampling. Due to the limited number of samples per year and non-equidistant sampling, the seasonal Mann-Kendall test was not applicable (Figure 2). Instead, significant seasonal patterns according to the Lomb-Scargle approach were subtracted prior trend analysis. The Mann-Kendall test was performed with the R-package 'Kendall' (McLeod, 2011).


### 3.4.3  Locally weighted regression (LOESS)

We assessed non-linear trends and low-frequency patterns with locally weighted regression (LOESS; Cleveland, 1979; Cleveland and Devlin, 1988), where the smoothing is done by local fitting of a second order polynomial to each point x in the

data set using weighted least squares. The weights for each value to be fitted are scaled to the range from 0 to 1 by the distance $d(x)$ between x and its $q^{th}$ closest point. The ratio of q to the number n of all data points, i.e. the span of the local regression smoother, defines the degree of smoothing. We used the default smoothing span which is a proportion of $q/n = 0.75$ of x´s nearest neighbours. Data

points further away than the $q^{th}$ data point do not contribute to the regression. Within the range of the span, the weights $w_i$ of the neighbouring points $x_i$ in the least square fit decrease with increasing distance of $xi$ to $x$ symmetrically around $x$ according to the tricubic weighting function $w_i(x) = (1 - [ (|x_i - x|) / d(x) ]^3)^3$. Again, significant seasonal patterns according to the Lomb-Scargle approach were subtracted prior

trend analysis. For details about choosing different LOESS-parametrisations, please see Cleveland (1979) as well as Cleveland and Devlin (1988). Local extrema of the LOESS smooth were identified with the R-package 'EMD' (Kim and Oh, 2009; 2014.).





## 4 Results

### 4.1 Multivariate components

We achieved the best performance of the Isomap dimension reduction with $k$ = 1300 (Table 2). In the following, results are presented for the first four Isomap components representing 88% of the interpoint distances of the total data set. For single sites (with more than 15 samples), between 29 and 97 % of the respective interpoint distances were represented (Table 2).

The 1$^{st}$ component depicted 44% of the interpoint distances of the total data set. Plotting residuals of the variables versus the 1$^{st}$ component showed strong positive correlations for $NO_3^-$, $Na^+$, $K^+$, $Mg^{2+}$, $Ca^{2+}$, $Cl^-$, EC, $SO_4^{2-}$, DOC and slightly less, but still positive, correlations for $O_2$ and Eh. Temperature was the only variable correlating negatively with the 1$^{st}$ component (Figure 3). Visualization of the component scores versus residuals of solute concentration revealed predominantly linear relationships (Figure S1).

The 2$^{nd}$ component reflected 19% of the interpoint distances in the data. It exhibited clear positive correlation with $O_2$ concentration, pH and Eh, and weaker correlation with $Na^+$, $K^+$ and DOC. It was inversely correlated with $Ca^{2+}$, EC and $SO_4^{2-}$ (Figure 3 and Figure S2). In the groundwater samples, $HCO_3^-$ and $Fe^{2+}$ had been determined as well. Both solutes were negatively correlated with this component (Figure 4 upper panel). $NO_3^-$ concentration in the deep groundwater samples was very low (with 27% of the samples below detection limit) and did not show any clear correlation with the 2$^{nd}$ component. Low component scores in the groundwater came along with high $Ca^{2+}$ and $HCO_3^-$ concentration.

The relationship of scores of component one and two in the groundwater is shown in the lower panel of Figure 4. Except for the two shallow wells close to the Peege stream (Gs_200, Gs_202; cf. Figure 1) scores of the 1$^{st}$ and 2$^{nd}$ component are negatively related (Figure 4 lower panel).

The 3$^{rd}$ component represented 6% of the interpoint distances in the data set. The residuals exhibited positive correlation for $Na^+$, $Mg^{2+}$, $Cl^-$, pH and temperature.





Negative correlations were found for $NO_3^-$, $Ca^{2+}$, $O_2$, Eh, and DOC (Figure 3 and Figure S3 ).

Another 22% of the interpoint distances in the data were assigned to the $4^{th}$ component. Residuals of the component scores showed negative correlation for $NH_4^+$, $PO_4^{3-}$, $K^+$, temperature, and DOC and positive correlation for $O_2$ (Figure 3 and Figure S4). The range of component values was spanned mainly by single large values of $NH_4^+$, $PO_4^{3-}$, and $K^+$ that cannot be explained with the preceding three

components (Figure S4). This highlights the importance of particular events for the $4^{th}$ component.

Table 2 Cumulated $R^2$ of the reproduction of the interpoint distances of the data in the projection by the first ten components of the best Isomap run and linear PCA.

| Component | 1 | 2 | 3 | 4 | 5 | 6 | 7 | 8 | 9 | 10 |
|---|---|---|---|---|---|---|---|---|---|---|
| Isomap | 0.42 | 0.6 | 0.66 | 0.88 | 0.94 | 0.96 | 0.97 | 0.98 | 0.98 | 0.99 |
| PCA | 0.39 | 0.57 | 0.65 | 0.88 | 0.94 | 0.95 | 0.97 | 0.98 | 0.99 | 0.99 |




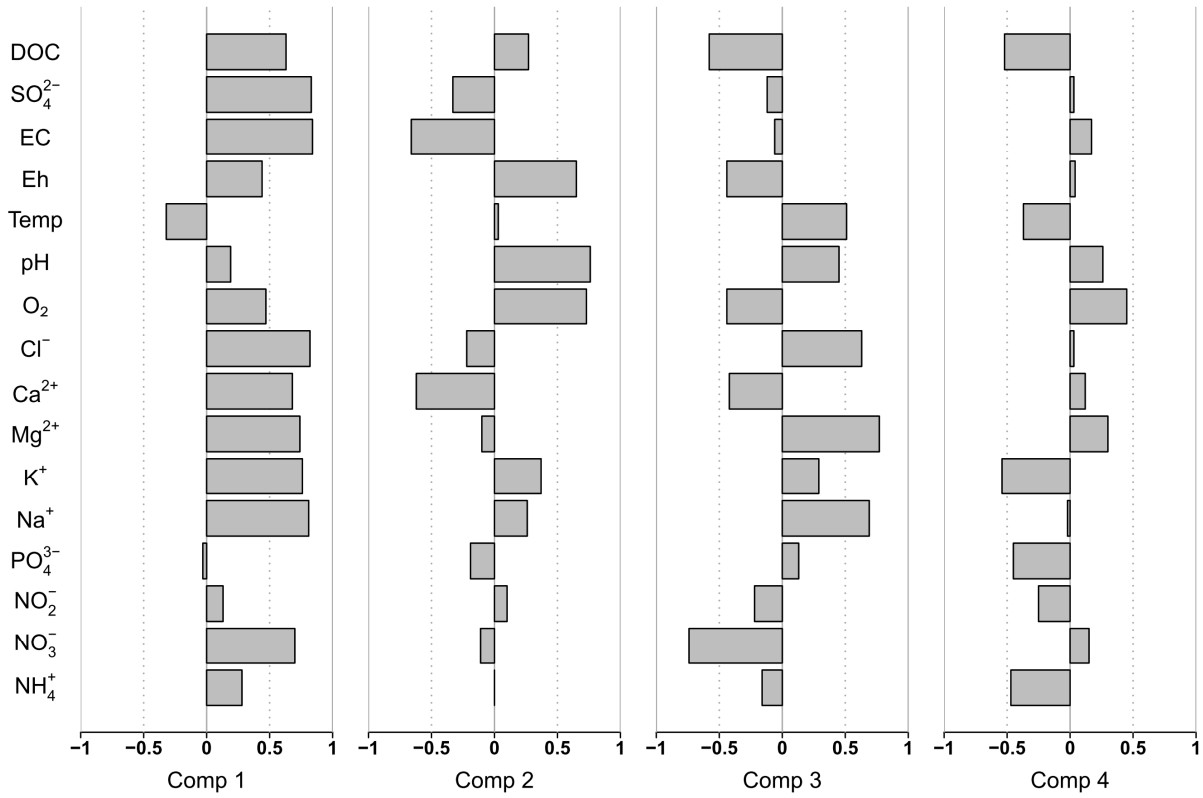

Figure 3 Spearman-rank-correlation of a component and the residuals of the multiple linear regression of the measured variable and the remaining three other components.

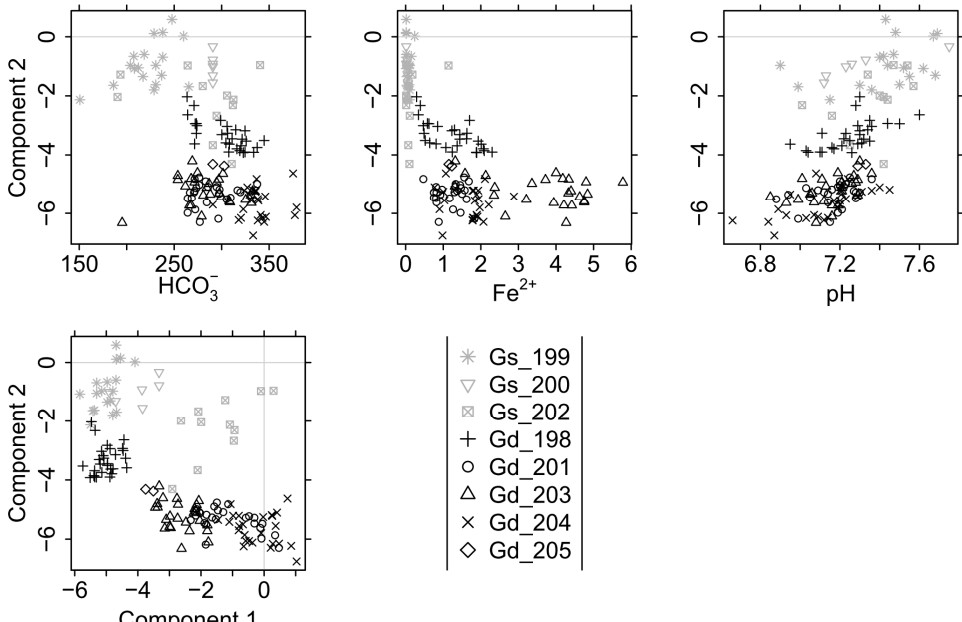

Figure 4 Upper panel: Selection of variables vs. scores of component 2 for the groundwater samples. Concentration in mgL$^{-1}$. Lower panel: Scores of component 1 vs. component 2 at the groundwater sites.

## 4.2 Multiple sites

Median values of scores of the 1$^{st}$ component clearly differed between streams (Figure 5 A). At the Strom sites, the median score values were considerably lower than those from the other stream water sampling sites. The median values of scores of the sites at the Quillow and Stierngraben showed intermediate values followed by the Ucker site, the Peege sites and finally the Dauergraben with the highest median score value. Groundwater samples in general exhibited consistently low scores of the 1$^{st}$ component, but without clear differences between deep and shallow groundwater samples. Mixing of water from different streams was visible at site Q_93 downstream the confluence of the Quillow (Q_95) and of the Strom stream (S_118), as well as at site Q_100 downstream the confluence of Q_104, Q_102 and P_107 (Figure 1 and





Figure 5 A).

Stream water samples exhibited the highest scores of the $2^{nd}$ component, whereas
low scores were limited to deep groundwater samples, and shallow groundwater
samples were in an intermediate position (Figure 5 B). Median values of the stream
water sites were approximately on the same level except for the sites Q_103, Q_106
and U_128 which exhibited noticeably higher median values than the other stream
water sites and the two Peege sites P_109 and P_108, which exhibited median
values on the same level as the shallow groundwater sites Gs_199 and G_200. The
scores in the deep groundwater clearly showed the largest absolute values,
indicating the significance of deep groundwater for this component (Figure 5 B).

Scores of the $3^{rd}$ component in the deep groundwater were consistently higher
than in shallow groundwater, while the stream water samples covered the whole
range of values (Figure 5 C). Lowest scores of the $3^{rd}$ component were found at the
Peege sites and in the shallow groundwater, highest scores at Ucker, Dauergraben
and the deep groundwater. At the Quillow stream, scores tended to increase from the
spring to the outlet. The effect of mixing of tributaries with different water qualities
was visible along the course of the Peege and Quillow streams downstream of the
respective confluences at the sites P_108, Q_95 and Q_93 (Figure 1 and Figure 5
C).

The range of values of the $4^{th}$ component was strongly biased towards negative
values, caused by single events at some sites which exhibited very low values
(Figure 5 D).





Figure 5 Boxplots of scores of component 1 to 4 at different sites. Sites with n < 13 are marked with '~', those with n < 3 with 2 '~'. Subscripts: P = Peege, Q = Quillow, S = Strom, St = Stierngraben, U = Ucker, D = Dauergraben, Gs = shallow groundwater, Gd = deep groundwater.

## 4.3 Long-term patterns

Time series of scores of the components were studied at sites with more than 50


temporal replicates. This applied for 13 stream water sites (Table S1). All dominant frequencies (for details, please see Appendix A) interpreted as seasonal patterns had a period length in the range between 350 and 380 days. For de-seasonalisation these seasonal patterns were subtracted from the time series prior to analysis for linear and non-linear trends.

Most of the time series of the scores of the $1^{st}$ component exhibited clear seasonal patterns with maximum scores during the winter season (Figure 6 and Figure 7). Between 30 and 67 % of the variance were assigned to the seasonal pattern. At all sites we found significant negative monotonic trends (Figure 6). The strongest decline was found at site D_112, the weakest trend at site Q_97 (not shown). The linear trend comprised between 9 and 48 % of the variance of the de-seasonalised time series (Figure 6). In contrast, the LOESS smooth depicted 14 to 57 % of the variance (Figure 6). It showed a decrease until December 2004 approximately and an increase thereafter (Figure 8). The de-seasonalised time series of groundwater heads showed a similar behaviour, with the minimum water level in June 2006 (Figure 8). Timing of the minimum values of the scores of the $1^{st}$ component varied between sites, spanning a range from $17^{th}$ February 2004 to $17^{th}$ of March 2009 (Figure 8). As an example, Figure 7 gives the time series of scores of the $1^{st}$ component at site Q_93, the seasonal pattern extracted from the series and the de-seasonalised time series with the non-linear trend estimated with the LOESS smoother.

Unlike for the $1^{st}$ component, only five of the thirteen considered time series of the $2^{nd}$ component exhibited a clear significant seasonal pattern, accounting for 17 to 48% of variance (Figure 6). The maxima of the seasonal patterns of the sites at Quillow and Ucker were in spring, at Stierngraben and Dauergraben in summer. In contrast, significant monotonic trends were found at most of the stream water sampling sites. All significant trends of the $2^{nd}$ component were positive. The linear trend comprised between 5 and 16 % of the variance of the time series, while the LOESS smooth comprised between 4 and 25 %.

Values of the $3^{rd}$ component showed a clear seasonal pattern with maxima in summer (Figure 6). Between 30 and 60 % of the variance were assigned to the



seasonal signal. The only exception was site D_112 were the seasonal pattern was

distorted by strong maxima in the winters of 2003, 2004 and 2007. Only at four sites significant linear trends were found. All of them were negative, comprising between 6 and 13% of the variance. The LOESS smooth depicted between 0 and 21 % of the variance.

For the 4th component, significant seasonal patterns with maxima in summer were

observed at 7 of the 13 analysed series, comprising between 17 and 61 % of the variance (Figure 6). Five sites showed a significant monotonic trend, comprising between 5 and 10 % of the variance. A negative trend was observed at site St_133 only. Four sites showed a positive trend. The LOESS smooth depicted between 1 and 16 % of the variance.






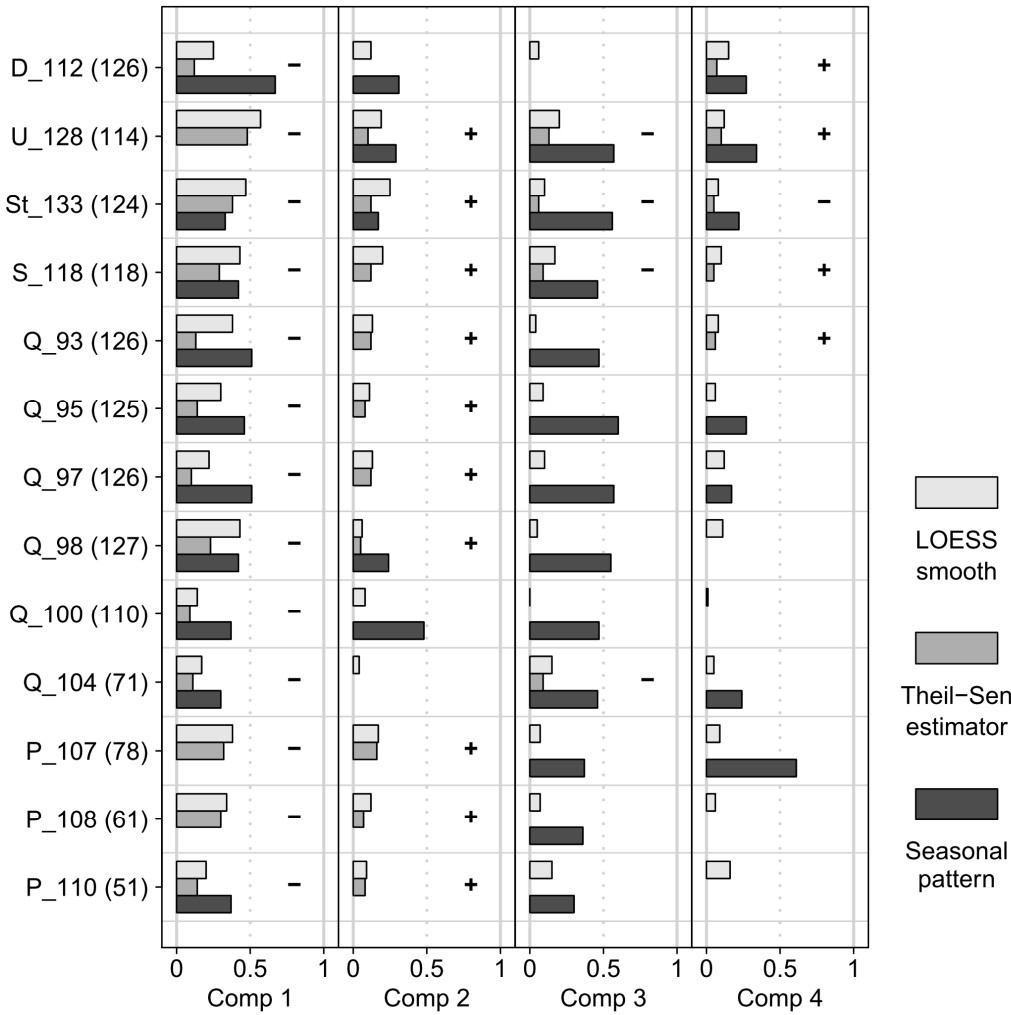

Figure 6 Fraction of variance of the time series of the Isomap component scores of sites with n > 50 assigned to the seasonal pattern (dark grey) and the trend estimated by the linear Theil-Sen estimator (mid grey) as well as the non-linear LOESS smooth (light grey). Fraction of variance is derived as $R^2$ of the scores of the respective component with the seasonal pattern or the estimated trend. Only significant seasonal patterns and linear trends are shown. The sign of the linear Theil-Sen estimator is given in the respective line. The number of samples at each site is given in brackets.




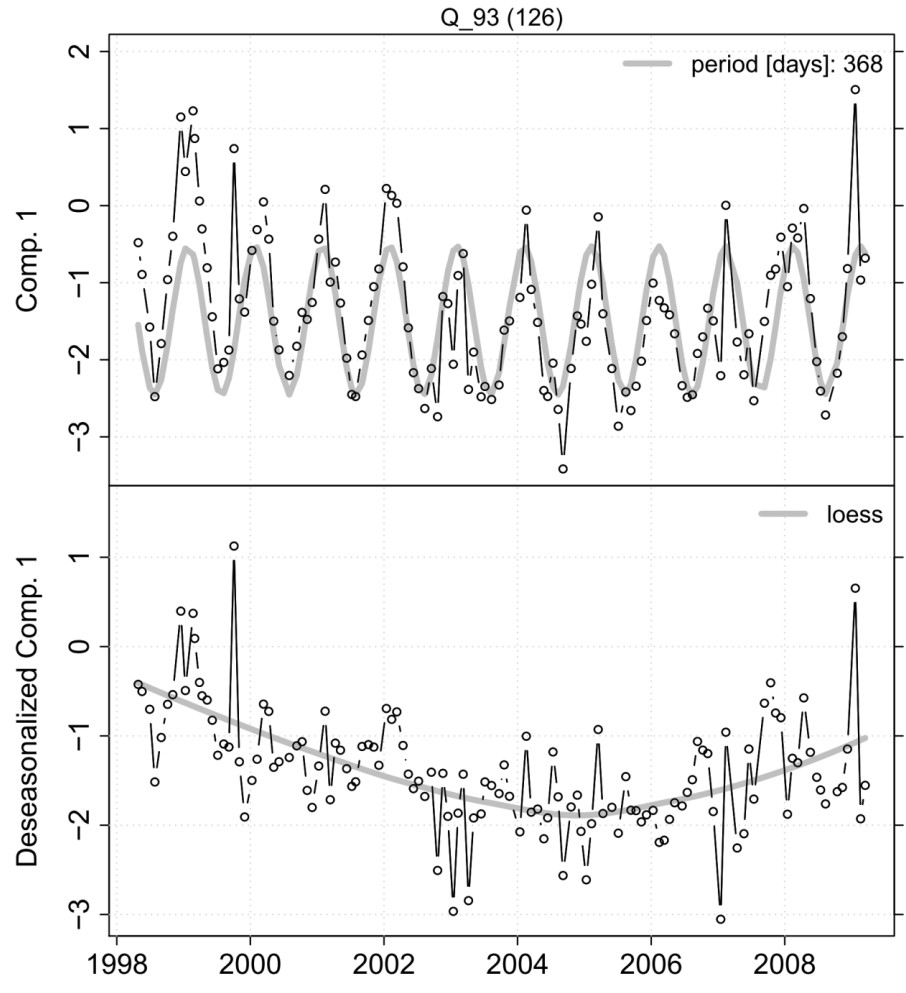

Figure 7 Upper panel: Time series of scores of the 1[st] component at site Q_93 in black and the seasonal pattern estimated with Lomb-Scargle in grey. Lower panel: The de-seasonalised series in black and the non-linear trend estimated with LOESS 600 in grey. The number of samples is given in brackets.

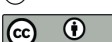

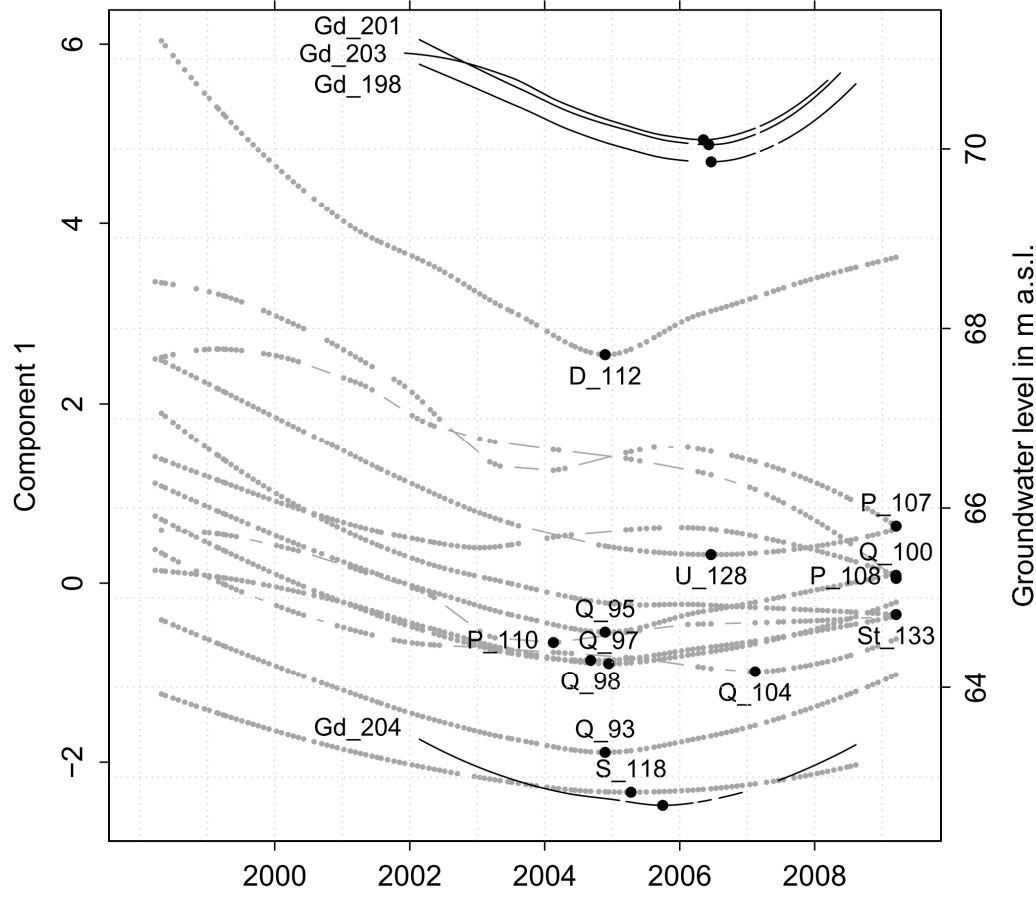

Figure 8 Left y-axis: LOESS smooth of time series of the 1$^{st}$ component at sites with n > 50 in grey. If a significant seasonal pattern was present, this was removed before smoothing. Right y-axis: LOESS smooth of the de-seasonalised groundwater level at four sites in black. The black dots mark the minima of the LOESS-smoothed series.





## 5 Discussion

### 5.1 Multivariate components

For PCA and Isomap, the 1st component represents by definition the correlation structure that predominantly can be extracted from the set of variables as a whole. If all the loadings of the 1st component of a PCA have the same sign, it is a weighted average of all the analysed variables (Jolliffe, 2002; Jolliffe and Cadima, 2016). The stronger the analysed variables are linearly correlated, the more the 1st component

approximates the arithmetic mean of all variables (for examples with hydrometric data see Lischeid, et al., 2010; Lehr et al., 2015). Furthermore, the 1st component serves as reference for all the subsequent components.

In this study each sample of the multivariate water quality data set is uniquely defined by a sampling site and a sampling date. Thus, the 1st component depicted a)

for each sampling site the pattern that was most prominent in the time series of the variables correlating with the 1st component, and b) between the sampling sites the difference in concentration level of those variables. High values of the 1st component indicate synchronous appearance of relatively high Eh and EC together with relatively high concentration of $NO_3^-$, $Cl^-$, $SO_4^{2-}$, $Na^+$, $K^+$, $Mg^{2+}$, $Ca^{2+}$, DOC, $O_2$ accompanied

with relatively low temperature (Figure 3).

In addition to the natural background, we assume a general effect of the agricultural practice on the solute concentration level and the dynamics of the water quality series in the area. Enhanced concentration of $NO_3^-$, $Cl^-$, $SO_4^{2-}$ and $Ca^{2+}$ is typical for groundwater and stream water in regions with intense agriculture

compared to forested areas (Broers and van der Grift, 2004; Fitzpatrick et al., 2007; Lischeid and Kalettka, 2012). Nitrogen and potassium are the main ingredients of mineral fertilizers. $Cl^-$ and $SO_4^{2-}$ are the dominating anions in potassium fertilizers. $SO_4^{2-}$ is a major ingredient of phosphorus fertilizers and ingredient in some nitrogen fertilizers. Calcite is present in some nitrogen fertilizers or is applied separately via

liming. DOC might be leached from slurry application or via tile drains after mechanical destruction of topsoil aggregates during tillage (Graeber et al., 2012). In addition, cations from the soil matrix might be leached by an enhanced anion




concentration (mainly $NO_3^-$) (Jessen et al., 2017). Overall the application of fertilizers and other agricultural practices like tillage tend to enhance the solute concentration

of seepage water (Pierson-Wickmann et al., 2009). Thus, we interpreted the 1st component as the enhancement of the natural background level of solute concentration due to agricultural practices.

Compared to the 1st component, the relationships of the 2nd component with Eh, pH and $O_2$ concentration were clearer expressed (Figure 3 and Figure S2). The

range of the scores of the 2nd component was spanned by the lowest values in the deep groundwater and the highest values in the stream water (Figure 5 B) whereas shallow groundwater exhibited intermediate scores. This sequence corresponds to redox conditions expected in those water categories. Thus, we interpret the 2nd component as a redox controlled component covering a sequence from reducing

conditions in deep groundwater to post oxic conditions in shallow groundwater and oxic conditions in stream water. $O_2$ and $NO_3^-$ concentration in deep groundwater samples usually was below the detection limit which is a common feature in this region (Merz et al., 2009). $NO_3^-$ in seepage and groundwater can be denitrified by microorganisms which use the oxidation of sulphides to sulphate as electron donor

for denitrification (Massmann et al., 2003, Jørgensen et al., 2009). We ascribed the high $SO_4^{2-}$ and $Fe^{2+}$ concentration to oxidation of pyrite (Figure 4 upper panel and Figure S2). Pyrite and other sulphides are abundant in the Pleistocenic sediments of North Germany (e.g. Weymann et al., 2010). Consequently, the pH decreases, calcite gets dissolved and the $HCO_3^-$ concentration increases. Part of the released

$Ca^{2+}$ replaces $Na^+$ and $K^+$ being sorbed to clay minerals.

We interpreted the clear separation in the 3rd component between relatively low scores for the shallow aquifer and relatively high scores for the deep aquifer as reflection of two opposing gradients (Figure 5 C). High concentration of $NO_3^-$, $O_2$ and DOC and relatively high Eh values being negatively related to the 3rd component

(Figure 3) is indicative for groundwater close to the surface, whereas enhanced concentration of the positively related solutes $Na^+$, $Mg^{2+}$ and $Cl^-$ is characteristic for local upwelling of saline groundwater from the underlying Tertiary aquifers at greater depth (Hannemann and Schirrmeister 1998; Tesmer et al., 2007). The scores of the




stream water samples, in turn, reflect the mixing ratio of groundwater from the two aquifers to the streamflow. We expect the baseflow maintained from the deep aquifer to be relatively enriched with geogenic solutes compared to the water that stems from the shallow aquifer or faster responding flow components like tile drain discharge and surface runoff. Water from the shallow aquifer is expected to be relatively enriched with surface born solutes compared to water that stems from the deep aquifer.

The range of values of the 4$^{th}$ component was dominated by single extremely low scores, reflecting samples with high concentration of $NH_4^+$, $PO_4^{3-}$, and $K^+$ (Figure S4). We conclude that these negative peaks can be ascribed to slurry application, being either unintentionally directly applied to the stream or being leached via surface runoff and tile drain discharge after application.


### 5.2 Multiple sites

The interpretation of the 1$^{st}$ component as agriculturally induced enhancement of the natural background level of most of the water quality variables is consistent with the spatial pattern of median component scores at the different sites. The highest 685 scores were found in the Dauergraben stream and in the Peege stream (Figure 5 A). Both catchments are characterized by intense agriculture, a relatively dense network of tile drains, and hardly any buffer strips along the streams leading to a rapid transmission of solute enriched waters from the fields to the streams. In contrast, the Strom stream exhibited the lowest scores among all streams. Compared to the other 690 streams, the valley of the Strom stream is clearly deep cut. Therefore, the Strom stream is expected to receive along its whole length continuous and substantial groundwater inflow from the deep aquifer. In addition, the valley slopes are covered with forest and not in agricultural use, acting as a buffer strip for the agricultural impact. Furthermore, the fraction of arable land in the Strom catchment is smallest, 695 and the fraction of woodland is largest compared to the other catchments (Table 1). Main parts of the Strom catchment are situated within a nature conservation area furthermore limiting the agricultural impact in its riparian zone.

Deep groundwater, shallow groundwater and the stream water were well




separated by the $2^{nd}$ component (Figure 5 B). Exceptions were the sites at the
Peege, which are mainly supplied with water from tile drainage and the shallow
aquifer and consequently yield median values similar to the shallow groundwater.
The largest positive median values of the $2^{nd}$ component, being higher than those of
the other stream water sites, were observed at sites with less than 13 samples
(Q_103 and Q_106) and at the site U_128 which received at least partly waters from
a different region than the other stream water sites (Figure 1 and Figure 5 B). Thus,
for the purpose of this study, we restricted our analysis on the spatial variability of the
redox component to the categories of deep groundwater, shallow groundwater and
stream water.

However, we took a closer look at the non-linear structure that became apparent
for the deep groundwater samples in some of the residual plots of the $2^{nd}$ component
(Figure S2). In addition, we related the groundwater values of the $2^{nd}$ component to
the $1^{st}$ component and the $HCO_3^-$ and $Fe^{2+}$ concentration (Figure 4). The negative
relationship between the $2^{nd}$ component and the $1^{st}$ component in the deep
groundwater suggests that the agricultural load represented by the $1^{st}$ component
acts as a driver for the sulphide oxidation represented by the $2^{nd}$ component. Among
all deep groundwater wells, the deepest groundwater well Gd_198 exhibited the
lowest scores of the $1^{st}$ component (Figure 5 A) and the highest scores of the $2^{nd}$
component (Figure 4 lower panel and Figure 5 B). This suggests that due to the
relatively low agricultural load the oxidation of sulphides was the least pronounced
among all deeper wells. Similar relationships between the extent of sulphate
oxidation in the aquifer and agriculturally borne $NO_3^-$ input have been found in other
studies (e.g., Zhang et al., 2009; Jessen et al., 2017 and references therein).

We expected the ratio of groundwater from the deep aquifer contributing to the
streamflow to increase in general with increasing catchment size. The Peege stream
is mainly fed by the shallow aquifer and yielded consequently median values of the
$3^{rd}$ component similar to the shallow groundwater sites (Figure 5 C). The streams of
Quillow, Strom and Stierngraben, showed little higher median values, indicating the
larger proportion of groundwater from the deep aquifer contributing to runoff
compared to the Peege stream. The sites U_128 and D_112 showed the highest





median values of the 3$^{rd}$ component among the stream water sites, being equal or
even higher than those of the deep groundwater sites (Figure 5 C). Both sites have
subsurface catchments that do not include the deep groundwater samplings sites in
this study. Site D_112 is on the eastern side of the river Ucker, while all groundwater
sampling sites are on the western side of it (Figure 1). In addition, its higher median

value of the 3$^{rd}$ component was partly due to several peaks during the winter time.
Those coincide with high values of Cl$^-$. These might indicate road salt application, but
we did not investigate this further, as it considered only this single site. Site U_128 is
situated at the outlet of the lake Unteruckersee upstream of the confluence of the
Quillow stream (Figure 1). There, we did not expect a contribution of the groundwater

sampled in the Quillow catchment either.

All the stream water sampling sites with negative peaks of the 4$^{th}$ component are
located near arable fields which are known to get fertilised by slurry (Figure 5 D). For
example the two most affected sites Q_102 and Q_103 receive slurry input from a
large pig farm close by (personal communication G. Verch). Overall, only a few slurry

input events accounted for 22% of the representation of the interpoint distances of all
the water quality samples of the water quality data set in the Isomap projection
(Figure 5 D). However, the percentage of represented interpoint distances of all
samples at a specific site ranged from < 1% to 42% for sites with n > 18 (Table S4).
This underlines the necessity to develop and use such methods in environmental

sciences which are able to consider non-linear processes and to deal with singular
and site-specific events.

### 5.3 Long-term patterns

Dominant changes were observed for the first two components (Figure 6). We
interpreted the non-linear long term trend of the 1$^{st}$ component at most stream water
sites (Figure 8) as the response of stream water quality to the interannual variability
of depth to groundwater. An increase in the thickness of the unsaturated zone leads
in general to longer residence time of seepage water, increasing retardation and
buffering of topsoil seepage water, which is reducing the solute concentration





originating from the surface in the seepage water and consequently reducing the values of the 1st component.

Trends similarly shaped to the non-linear trend of the 1st component of stream water quality were observed for the water level in the deep groundwater. In general, the turning points of the deep groundwater head time series lag behind those of the

scores of the 1st component of the stream water sites by approximately 1.5 years (Figure 8). The earlier date of the turning point at groundwater gauge Gd_204 in October 2005 is most probably an artefact, caused by the effect of the large time gaps in 2006 and 2007 on the de-seasonalising at this site and has to be considered with care.

We suggest that the time lag between stream water chemistry and water level in the deep aquifer is due to different response times to changes in the moisture conditions of the unsaturated zone. Compared to the relatively fast response of the stream water quality, the groundwater level in the deep aquifer reacts slower. In general, the overall trend of groundwater recharge reflects a relatively slow response

to changes in the regional water balance. The velocity of seepage in the sediments of the upstream region of the Quillow catchment is estimated to be roughly 1 m per year.

The seasonal patterns, i.e. the annual variability, in the time series of the scores of the 1st component in the streams were ascribed to transient hydraulic decoupling of

the mostly affected topsoils from the streams in summer. Usually there is hardly any seepage during the dry summer months at all. This leads often to desiccation of the uppermost stream reaches (Lischeid et al., 2017). Thus, shallow groundwater and tile drain discharge, both sources with relatively high agricultural load, did not contribute to stream discharge during these periods and larger areas of the catchment got

hydraulically decoupled from the stream network (Merz and Steidl, 2015). Similar effects of the seasonal variability of the hydrological connectivity of streams, groundwater and tile drainage in agricultural catchments on the concentration level of agriculturally born solutes in the stream water have been reported, e.g. for $NO_3^-$ in the Schaugraben study catchment in the North of Germany (Wriedt et al., 2007) and

for $NO_3^-$ and $Cl^-$ in the Kervidy-Naizin catchment in western France (Molenat et al.,



2008; Aubert et al., 2013).

The other dominant change of stream water chemistry observed in this study was the continuous increase of the 2nd component at most stream water sites (Figure 6). All of the sampling sites with very low values of the 2nd component were in the deep

aquifer (Figure 5 B). The positive trends of the 2nd component at most stream water sites were ascribed to changes in the chemistry of the groundwater-borne baseflow. Considering the interpretation of the 2nd component, this translates into enhanced oxidation of geogenic sulphides in the deeper aquifer due to the continuous input of agriculturally born $NO_3^-$ and DOC and subsequent calcite dissolution. Geogenic

sulphides, such as pyrite, serve as electron donors for denitrification. The consumption of the geogenic sulphides is irreversible and might lead to the depletion of the denitrification capacity in the deep aquifer in the long run (Merz et al., 2009; Zhang et al., 2009; Merz and Steidl, 2015). Consequently, buffering of $NO_3^-$ surplus from agricultural land use is expected to decrease and $NO_3^-$ concentration in the

groundwater and the stream water is expected to increase. The hypothesised long-term development should be of concern for the water resources and environmental protection agencies with respect to future water quality and related international commitments, such as the Water framework (EU, 2000), the Groundwater (EU, 2006) and the Nitrate directive (EU, 1991) of the European Union. Substantial time lags

have to be considered for the response of groundwater quality to measures that reduce leaching of $NO_3^-$ (e.g. Pierson-Wickmann et al., 2009; Meals et al., 2010). In the Quillow catchment, we expect travel times in the order of magnitude of decades for the seepage water to reach the deep aquifer.

We did not observe dominant changes for the other two water quality components

during the course of the observation period. The main temporal feature of the 3rd component was a very distinct and steady seasonal pattern, as could be expected for the mixing ratio of groundwater from the deep aquifer. All stream water sites with n > 50, except for D_112, showed a distinct seasonal pattern with maximum scores in the summer, which is consistent with the assumption that the fraction of deep

groundwater in the streams is highest during this period (Figure 6). The seasonal pattern at site D_112 was disturbed by the winter peaks we ascribed to road salt





application (section 5.2).

Because of its strong dependence from single events (Figure 5 D), the results of
the estimation of the seasonal patterns and the trends of the 4th component have to
be considered with care. The maxima of the seasonal pattern in summer at some
sites were interpreted as reduced nutrient inputs to the stream due to nutrient uptake
of plants and maximum buffering capacity of the unsaturated zone in summer. There
were no indications for any effects of those events that we ascribed to the direct
effect of slurry application on samples taken on the subsequent sampling dates at the
affected sites. This is presumably due to the width of the sampling interval (Figure 2).

In case of dependence of a component from single events, 'change' might be also
related to clustering of those events during certain parts of the series, either for series
at single sites or sets of series. Most of the 'extreme' events of the 4th component
appeared during the first half of the observation period (not shown). However,
because of the small number of clearly outstanding events, we did not investigate this
further (Figure 5 D).

In this study, the presented analysis of changes in water quality was limited by the
temporal resolution of the data. Aspects such as long-term memory effects, as
indicated by fractal scaling of solute series (Kirchner et al., 2000) and the observed
scale-crossing non-self-averaging behaviour of solute series (Kirchner and Neal,
2013) were not considered. However, we assume that the suggested use of
multivariate components gives some robustness to the detected changes compared
to the analysis of single solutes.

**5.4 Effects of the irregular sampling**

It is important to note that our approach does not require identical temporal
sampling resolution at all sites (Figure 2). Occasional sampling at additional sites
helps assessing the strength of effects of the respective drivers at these sites and
might support or contradict hypotheses on spatial variability and related long-term
patterns of those influences. In addition, the approach followed here does not require



synchronous sampling dates. Thus, a strictly regular sampling design, which is hardly feasible, is no prerequisite. Correspondingly, data from different monitoring programs could be used for a joint analysis.

Sampling intervals at the sampling sites with N > 50 were not normally distributed and biased towards deviations that are longer than the median (right panel Figure 2). Several series exhibited large time gaps. However, as sampling intervals did not change systematically throughout the monitoring period we assume that the effects on the results of the significance test with Mann-Kendall were negligible (section 3.2). In comparison, the trend estimations with Theil-Sen estimator and LOESS are more

robust, as they incorporate the exact sampling dates explicitly in the calculations. Thus, we do not expect major effects on the sign of the Theil-Sen estimator or the general shape of the LOESS smooth at the given temporal resolution.

There was an obvious spatial bias with a focus on the Quillow catchment itself, conditioned by the focus of the monitoring (section 2.2, Figure 1). Stream sampling

sites were only partly independent from each other, as the same streams had been sampled along different reaches. This needs to be considered in the interpretation of the components. In our exploratory approach, differences between subsequent stream reaches helped to identify the effects of tributaries or groundwater that recharged between the respective sampling sites. In that way, the stream was used

as a measurement device for biogeochemical processes and water-borne solute transport in different parts of the catchment and the interlinkages of groundwater and stream water.

### 5.5 Exploratory framework

The application of a dimension reduction approach was motivated by the assumption that drivers influencing water quality usually affect more than one solute, and that single solutes are affected by more than one driver. Like in preceding studies (e.g., Lischeid and Bittersohl, 2008; Lischeid et al., 2010), the representation of water quality data in low-dimensional space required only a few components to

capture the 'main features' of the data set. Non-linear Isometric Feature Mapping





performed in this study only slightly better with respect to the representation of interpoint distances than PCA (Table 2), suggesting that mainly linear relationships were of importance for the overall dynamics in the data set.

This is usually not known in advance. Thus, if the aim is to consider and check for 885 possible non-linear relationships in the analysis we recommend using PCA as a linear benchmark for Isomap (demonstrated by Lischeid and Bittersohl, 2008). In a straightforward way this allows for 1) assessing whether the dominant correlation structures in the data set are mainly linear or non-linear, and 2) identifying those components, samples, sites and periods deviating from the linear behaviour as 890 captured by the PCA.

Based on the correlation of component scores and residuals, we formulated for each considered component a hypothesis on a dominant driver influencing water quality. The assessment of the relationships of scores and residuals with Spearman rank-correlation considers non-linear monotonic correlations and is less sensitive to 895 extreme values compared to Pearson correlation. The derived correlations differ from default loadings of PCA, which are defined as the coefficients of the linear combination of the analysed variables which is used to calculate the principal component scores. Those coefficients, scaled by the square root of the eigenvalue of the respective component, are equivalent to the Pearson correlation of PCA 900 component scores and analysed variables. It is important to note that the differences in the evaluation of the correlations of components and the measured variables might lead to different interpretations of the components.

Complementary, we used for the derivation of the hypotheses the spatiotemporal features of the components in combination with the spatial order of the sampling 905 sites, other variables like groundwater level series, $Fe_2^+$ and $HCO_3^-$ concentration from the groundwater samples, the spatial distribution of land use, and expert knowledge on the study area. A thorough testing of the hypotheses, for example through hydrochemical modelling or numerical experiments with virtual catchments was out of the scope of this study.

However, an interpretation of the components as distinct drivers is no prerequisite



for the further analysis of the components. In any case, the components constitute, and can simply be used as, a condensed representation of similar behaviour among the analysed variables according to the constraints of the used dimension reduction method.

For PCA and Isomap each component describes subsequently the correlation structure that is most prominent in the remainder of what has not already been assigned to the higher-ranked components. This implies that each component has to be interpreted with respect to the higher ranked components. Also, the consideration of the respective other components in the interpretation of a component can be

helpful to carve out its characteristics as it was done here with the residuals of the multiple linear regression of the respective three other components and the measured variables (e.g. Figure S1). Beyond that, it can be helpful to elucidate the interaction of the components as it was done here e.g. for scores of the $1^{st}$ and $2^{nd}$ component (Figure 4 lower panel).

The sites differed substantially with respect to the median values of the four analysed multivariate components (Figure 5). However, these components comprised the largest fraction of the interpoint distances at any single site with more than 18 samples (Table S4). We conclude that our results identified major regional phenomena rather than site-specific peculiarities. This is consistent with the prior

assumptions that the dominant drivers influencing stream water and groundwater quality were in fact the same at all sites. This gives some confidence to hypothesize that these drivers presumably play a major role even in adjacent catchments that have not been sampled so far.

To detect and characterize the dominant changes in the multivariate water quality

data we explored whether there were shifts in time in specific components, whether they were linear or non-linear in nature, and if trends did occur at many or only at single sites. For example for the scores of the $1^{st}$ component, the Mann-Kendall approach identified monotonic trends at various stream water sampling sites (Figure 6). However, the linear trend estimation failed to detect the non-linear trend that was

observed at many series (Figure 8). This reflects the well-known sensitivity of global linear trend estimation to low-frequency patterns that are not entirely covered by the



observation period (Koutsoyiannis, 2006; Milliman et al., 2008; Lins and Cohn, 2011).

The LOESS smooths of the de-seasonalised series, on the other hand, did clearly reveal the similarity between the long term behaviour of groundwater level in the deep aquifer and series of the 1st component. In our exploratory approach, the LOESS smooth of the de-seasonalised series served as a descriptive tool for illustrating rather than for proving non-linear long-term patterns. No significance test was applied. The outcome of the LOESS smoother highly depends on the parameterisation of the approach (i.e., the degree of smoothness) that would have to be justified prior testing of significance.

## 6  Conclusions

We suggested and tested an exploratory approach for the detection of dominant changes in multivariate water quality data sets with irregular sampling in space and time. The combination of the selected methods aimed to provide a broadly applicable exploratory framework for typical existing monitoring data sets, e.g. from environmental agencies, which are often characterized by relatively low sampling frequency and irregularities of the sampling in space and / or time. In the approach, we applied a dimension reduction method to derive multivariate water quality components and analysed their spatiotemporal features with respect to changes that concerned more than single sites, short-term fluctuations or single events.

The components can be used irrespective of an interpretation as drivers influencing water quality. By definition, the components are a sparse description of the common dynamics among the water quality variables. Thus, similar behaviour in space and time among the water quality variables as well as systematic changes in the multivariate water quality data can be addressed in a purely descriptive manner. This can be used for example to test the often implicit assumption of constant boundary conditions of scientific process and modelling studies. Furthermore, the components and their spatiotemporal features per se can serve as reference for further studies, e.g. detailed process studies with higher temporal resolution, and the assessment of future developments of water quality in an area. In this study, the





components were used to develop hypotheses on dominant drivers influencing water quality and to analyse the temporal and spatial variability of those influences.

It is emphasized that the presented approach is readily applicable with data from common monitoring programs without specific requirements concerning sampling frequency or regular distribution of sampling sites, sampling dates, and sampling intervals, except that there should be no systematic bias in the respective distribution. Even variables which have to be excluded from the derivation of the components, for example because of the amount of missing values or because they have been monitored only at subsets of the sampling sites, can be related to the components as additional information for their interpretation. For example in this study we used the concentration of $Fe^{2+}$ and $HCO_3^-$ in the groundwater as additional information for the interpretation of the $2^{nd}$ component. Thus the approach allows an efficient use of existing monitoring data as well as the consideration of often neglected 'irregular' pieces of data from e.g. pilot studies or single sampling campaigns. Irregularities in the structure of a data set are not seen as fundamental hindrance, but as additional source of information. We see this as a major advantage for the analysis of comprehensive water quality monitoring programs, both from a scientific perspective and from a more applied point of view of e.g. water resources and environmental agencies. Therefore, we recommend the approach especially for the exploratory assessment of existing long term low frequency multivariate water quality monitoring data sets.

**Data availability**

A selection of R-scripts that covers the main steps of the exploratory framework is provided at http://open-research-data.ext.zalf.de/ResearchData/2017_340.html under CC-BY 4.0 licence. It comes together with the water quality data used in this manuscript and some examples of exploratory plots not included in this manuscript.



## Acknowledgements

The long-term monitoring program that provided the data for this study would not have been possible without the diligent work of many colleagues. We would like to thank Roswitha Schulz, Dorith Henning, Ralph Tauschke, Joachim Bartelt (†), Peter Bernd and Bernd Schwien for installation of sampling sites, including numerous groundwater wells, and for performing the sampling program in spite of sometimes harsh field conditions. In addition we acknowledge the painstaking work of Rita Schwarz (†) and Melitta Engel in the chemical laboratory of the Institute of Landscape Hydrology as well as of the staff of the central chemical laboratory of the Leibniz Centre for Agricultural Landscape Research. We thank Gernot Verch of the research station Dedelow for the information on the historical development of agricultural land use in the study area.

Christian Lehr received funding from the Leibniz Association (SAW-2012-IGB-4167) within the International Leibniz Research School: Aquatic boundaries and linkages in a changing environment (Aqualink).

We used free software products under the GNU General Public Licence and thank the respective communities. Maps and the determination of the catchments' areas were carried out with Quantum GIS 2.14.1 (http://www.qgis.org/index.php) and statistical analyses and the graphs were performed using the R statistical software environment, version 3.4.1 (R Core Team, 2017; http://www.r-project.org).





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





**Appendix A**

**Lomb-Scargle**

A given discrete time series $Y(t_i)$ with $(i = 1,...,N)$ and centred around zero can be

described as a superposition from sin- and cos-terms with amplitudes *a* and *b*, time

$t_i$ , angular frequency $\omega = 2\pi f$ and a noise term $n_i$ .

$$Y(t_i) = a\cos\omega t_i + b\sin\omega t_i + n_i \qquad (1)$$

Lomb (1976) introduced an additional factor Tau to consider for deviations from the

evenly spaced case.

$$\tau_j = \frac{1}{2\omega_j} \bullet \arctan 2\left( \sum_i^N \sin 2\omega_j(t_i - t_{ave}), \sum_i^N \cos 2\omega_j(t_i - t_{ave}) \right) \qquad (2)$$

The constant $t_{ave} = min(t) - max(t)$ scales the term to the centre of the period covered

by the series for every frequency *j*. If the starting point of the series is set to zero $t_{ave}$

enables to correct for offsets between the spectral components and thus allows to

correctly reconstruct the original series out of its spectral components (Hocke 1998;

Hocke and Kämpfer, 2009).

With these two extensions of the time term, equation 1 can be rewritten as

$$Y(t_i) = A\cos(\omega(t_i - \tau - t_{ave}) + \phi) + n_i \qquad (3)$$

with amplitude $A = \sqrt{a^2 + b^2}$ and phase $\phi = \arctan(b/a)$.

The Lomb-Scargle periodogram $P_N(\omega)$ (equation 4) normalized with the total variance

of the data $\sigma^2$ equals the linear least square fit of the time series model in equations

1 and 3 for a certain frequency (Lomb, 1976; Press et al., 2007).

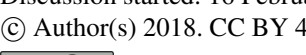



$$P_N(\omega) = \frac{1}{2\sigma^2} \left\{ \frac{\left( \sum_i^N Y(t_i) \cos[\omega_j(t_i - \tau - t_{ave})] \right)^2}{\sum_i^N \cos^2[\omega_j(t_i - \tau - t_{ave})]} + \frac{\left( \sum_i^N Y(t_i) \sin[\omega_j(t_i - \tau - t_{ave})] \right)^2}{\sum_i^N \sin^2[\omega_j(t_i - \tau - t_{ave})]} \right\} \quad (4)$$

The amplitudes *a* and *b* can be computed out of the square root of the corresponding
sin- and cos-terms of the normalized Lomb-Scargle periodogram, which yields the
normalized power spectral density at certain frequencies (Hocke and Kämpfer, 2009).

$$a = \sqrt{\frac{2}{N}} \frac{\sum_i^N Y(t_i) \cos[\omega_j(t_i - \tau - t_{ave})]}{\sqrt{\sum_i^N \cos^2[\omega_j(t_i - \tau - t_{ave})]}} \quad b = \sqrt{\frac{2}{N}} \frac{\sum_i^N Y(t_i) \sin[\omega_j(t_i - \tau - t_{ave})]}{\sqrt{\sum_i^N \sin^2[\omega_j(t_i - \tau - t_{ave})]}} \quad (5)$$

Different modified series can be reconstructed out of any set of spectral components.
So the method might be used i.e. as band-pass-filter or filling of gaps in the series
(Hocke and Kämpfer, 2009).

The number of frequencies in which the series is decomposed is calculated with the
empirical formula derived out of Monte Carlo simulations by Horne and Baliunas
(1986) (Glynn et al., 2006; Press et al., 2007).

$$N_{indep} \approx -6.362 + 1.193N + 0.00098N^2 \quad (6)$$

The false-alarm probability or statistical significance level $p$ of the $P_N(\omega)$ value at a
certain frequency is calculated with equation (Scargle, 1982; Glynn et al., 2006;
Press et al., 2007).

$$p = 1 - \left(1 - e^{-z}\right)^M \quad (7)$$

$M$ is the number of test frequencies which is here set to $N_{indep}$ and *z* is the tested
value of $P_N(\omega)$ at a certain frequency. To diminish aliasing, which means reappearing
of higher frequencies' power in the power of lower ones, the highest test frequency is
set to the Nyquist-rate $f_{max} = f_{Nyquist} = 1/(2\Delta t)$. Because of the irregular sampling, the

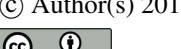



sampling rate $\Delta t$ is approximated here by the average sampling interval $\Delta t = (t_N - t_i)/N$. The lowest test frequency is the inverse of the sampling range

$f_{min} = 1/(t_N - t_1)$ (Scargle, 1982; Press et al., 2007).

Although $N_{indep}$ should be the number of independent frequencies in the signal it is possible that frequencies lying close to each other 'share' the same underlying trigger. This leakage of power is promoted by the uneven sampling and oversampling of the frequency domain *M > N* (Scargle, 1989; Horne and Baliunas, 1986). In

addition, the effect may be enhanced because of local high sampling density, autocorrelation in the data or very strong momentum of the underlying trigger.

With regard to these circumstances, which apply especially for the groundwater level series in this study, only the 'dominant' frequencies were used to identify seasonal patterns. The term 'dominant' frequency is used here for the peaks in between

1385 groups of significant frequencies. If such groups build 'significance-plateaus' the median of this plateau is taken as dominant frequency.