# Peer review of "Detecting dominant changes in irregularly sampled multivariate water quality data sets"

_Hydrology and Earth System Sciences, 2018_

## Referee Comment (RC1) · Anonymous Referee #1 · 16 Mar 2018

This manuscript presents a new exploratory framework for detection of dominant changes in multivariate water quality data sets with irregular sampling in space and time. The paper is well written and I think it is a valuable contribution to the hydrological community. I recommend its publication after the following comments are addressed.

General comments:

1. On the novelty of the proposed framework: I think this manuscript can foster future research ideas and efforts that are aimed toward detecting dominant changes in watershed using multivariate data at multiple sites. I think this type of coherent and systematic investigation of watershed data is limited in the literature, since previous studies have tended to focus on either only a few sites or a few constituents.

2. On the abstract: I found it quite lengthy (469 words), which prevents readers from quickly grasping the key messages. Also, it is not customary to have more than one paragraph in the abstract.

3. On the coverage of the monitoring data: The paper addresses the 'time' aspect of the collected water quality data but lacks a thorough discussion on the 'discharge' and 'season' aspects of those data. Were all constituents at these sites sampled roughly similarly across season? Were they sampled roughly similarly during normal-flow and stormflow conditions? Such information is important and can be simply shown with boxplots (e.g., with "month" and "discharge percentiles" as x-axes respectively.) If samples at these sites were not taken roughly similarly across season or discharge, how would that affect the validity of the proposed exploratory framework and the interpretation of the results? The authors should comment on that.

4. On the general applicability of the framework: Several points shall be discussed by the authors regarding the applicability of the framework, which can guide its application to monitoring network elsewhere. a) Is the framework intended to solute data only? Sediment and total phosphorus are typically monitored by many programs. Do the authors recommend the inclusion of such constituents in the proposed framework? b) What is the threshold for a constituent (or a site) to be included in the analysis? Specifically, how many samples are required for a constituent-site pair to be eligible? I am puzzled by the few stations in Figure 2 that have only 1-8 samples. I wonder whether these site-constituent pairs should be disregarded. c) For such multi-site and multi-constituent exploration, all available data should be considered to enhance the robustness of modeling results. However, not all the data are consistently available across the sites. Then, how should one handle the tradeoff between the number of constituents and the number of sites? If we rank all constituents by the number of applicable sites, C1, C2, C3, C3, …., C16, then what is the relative gain of sequentially adding extra constituents (from C1 to C16) into the analysis framework? Can an explicit rule be developed to prevent adding new constituents to the framework?

5. On the irregularity nature of the monitoring data: The authors have provided adequate references in many parts of the manuscript. One exception is on the irregularity of water quality data ($\sim$ line 110 and also Section 5.4). One reference that you may find useful is provided below, which discusses at least two points that are discussed in this manuscript, including (a) irregularity nature of water quality data and how to model that property and (b) fractal scaling in water quality data which may affect trend significance (including the trend approaches used here).

Zhang, Q., Harman, C. J., and Kirchner, J. W. (2018), Evaluation of statistical methods for quantifying fractal scaling in water-quality time series with irregular sampling, Hydrol. Earth Syst. Sci., 22, 1175-1192, https://doi.org/10.5194/hess-22-1175-2018.

Specific comments:

6. On Figure 2: a) This is a well-designed figure. b) Consider adding vertical reference lines in the right panel to indicate 1-day, 1-week, and 1-month intervals. c) Add additional reference lines to separate groundwater from stream water – refer to your treatment in Figure 5. d) Consider using color to distinguish between median and mean. e) Comment in the text on the apparent outlier in the site GdQ_198 distribution. f) Do the numbers in bracket represent the number of samples for one constituent or all constituents? Clarify. g) Two of the sites have only one sample each. Justify why those sites should not be removed. In my opinion, those sites which only several samples should also be excluded unless their use can be justified.

7. Line 248: I would suggest using median for the missing value replacement.

8. Line 252: Provide references to justify the use of half detection limit for censored values. It is a typical practice but it has been pointed out that such treatment may cause issues to analysis – refer to the references below. This could be a problem for NO2 and PO4, since the two species have significant proportions of censored values (Table S3).
Helsel, D.R., 2006. Fabricating data: how substituting values for nondetects can ruin results, and what can be done about it. Chemosphere, 65(11), pp.2434-2439.

Helsel, D. R. (2005). More than obvious: better methods for interpreting nondetect data. https://pubs.acs.org/doi/pdf/10.1021/es053368a.

9. Line 262: How was the threshold of '50 samples' chosen? It is still a small size.

10. Line 386 (Eq. 2): Check whether you want to use two equal signs in this equation.

11. Line 421: The effect of autocorrelation on trend analysis is not only relevant to short-memory processes (e.g., AR(1) in Yue et al., 2002), but also long-memory processes (e.g., ARFIMA).

Cohn, T. A., and H. F. Lins (2005), Nature's style: Naturally trendy, Geophys. Res. Lett., 32, L23402, doi:10.1029/2005GL024476.

Zhang, Q., Harman, C. J., and Kirchner, J. W. (2018), Evaluation of statistical methods for quantifying fractal scaling in water-quality time series with irregular sampling, Hydrol. Earth Syst. Sci., 22, 1175-1192, https://doi.org/10.5194/hess-22-1175-2018.

12. Line 456: I think it should be 42% (per Table 2).

13. Line 459: In addition to temperature, $PO_4$ is also negatively correlated with PC 1.

14. Line 463: This should be 18% (per Table 2).

15. Line 537: Check the label for $n < 3$ in Figure 5, which should not be identical to $n < 13$.

16. Line 675: This conclusion should be supported by some references.
* * *

---

## Referee Comment (RC2) · Anonymous Referee #2 · 22 Mar 2018

The manuscript proposes an exploratory framework for detection of dominant changes in multivariate water-quality data sets with irregular sampling in space and time. As stated in the introduction, many analysis methods assume regular temporal spacing, but many monitoring networks evolve over time resulting in irregularly spaced samples. The concept is good, some more effort needs to be put into the writing and analysis.

1. The abstract is rather lengthy.

2. The introduction contains vague statements and extraneous adverbs. The first sentence of the article is "Numerous high frequency studies unravelled the high temporal variability of stream water quality." This is well known, as shown by the many references. It seems like the first sentence of the article should start with a stronger sentence about the problem at hand. The second paragraph of the introduction has the

phrase "numerous different drivers at different scales." This is vague. Give an example, or qualify the drivers, such as climatic and land-use drivers. The second sentence of the third paragraph is either missing something or "determining" should be "determine."

3. In the description of the study area the mean annual precipitation and mean annual temperature are given for the federal state Brandenburg for 1961–1990. This does not overlap with the study period of 1990–2009 at all. With the common use and availability of climatic data, it would not take much effort to report precipitation and temperature for the study period. It is not clear what period the water balance variability values represent.

4. The topography and soils sections are well written and informative.

5. We know the data are collected irregularly, but are they collected to be representative of seasons and flow conditions, i.e., are there high-flow samples?

6. Figure 2 shows some sites with very little data, yet it seems like they were included. It is not clear how these help inform the method. It seems like there should be some minimum number of samples per year most of the years from 1998—2009 in order for a site to be included in the study. Some parts of the proposed framework were done for sites with more than 50 observations. It seems like the entire analysis should be done only with those sites. It is not clear how these low-sample sites fit with the rest of the sites.

7. It has been very well documented that substituting a fraction of the reporting limit is an inappropriate method for dealing with censored data. See:

Gilliom, R.J., and Helsel, D.R., 1986, Estimation of distributional parameters for censored trace level water quality data, 1. Estimation techniques: Water Resources Reserach, 22, 135–146.

Singh, A., and Nocerino, J., 2002, Robust estimation of mean and variance using environmental data sets with below detection limit observations: Chemometrics and Intelligent Laboratory Systems, 60, 69–86.

Helsel, D. R., 2005, More than obvious—Better method better methods for interpreting nondetect data: Environ. Sci. Technol., 39(20), 419A–423A, DOI: 10.1021/es053368a

Helsel, D.R., 2005, Nondetects and Data Analysis: Wiley-Interscience, 250 p.

Helsel, D.R., 2006, Fabricating data—How substituting values for nondetects can ruin results, and what can be done about it: Chemosphere, 65(11), 2434–2439.

Helsel, D.R., 2012, Statistics for Censored Environmental Data Using Minitab and R: John Wiley & Sons, 324 p.

Admittedly, the percent of censored values is small, but substitution should really not be used anymore in water-quality analyses. I'm not sure if Isometric Feature Mapping can utilize censored values. However, the authors could estimate the mean and standard deviation of the constituents with censored values using regression on order statistics or maximum likelihood methods (see Helsel, 2012) before standardizing the variables. The Akritas-Thiel-Sen median line can be used for the trend analysis.

8. Check equation (2) in line 385. Should there be a plus sign between B0 and the summation symbol? Describe the components of the equation that were not already described in equation (1).

9. In the interpretation of components, the authors describe using multiple linear regression, which is a parametric method that assumes a model linear in the parameters, but then make an argument for a non-parametric measure of correlation applied to the multiple linear regression results. This seems contradictory.

10. Consider presenting the methods and the results in the same order for parallel construction.

11. In the discussion, the conclusions on page 32 about the 1st component were not

well supported. There were a lot of statements like "we assume a general effect," some process "might" happen, some processes "tend to enhance." The discussion of the 2nd component was better supported with information about the sediments in the area. Some of the material in the first paragraph of section 5.2 should be moved up to better support the conclusions about the 1st component. The discussion of the 4th component on page 33 seemed speculative. Has this been modelled or shown elsewhere?

12. Page 37 states nicely some important implications of the observed water quality.

13. Page 40, line 895, change "is" to "are."

14. Page 40, line 901, "Complementary" does not seem like an appropriate word for this sentence.

15. Some of the results, discussion, and conclusions mention both PCA and Isomap, but some of the numbers, figures, results must come from one of them specifical-ly—that should be made more clear.

16. Check that numbers in the text agree with the numbers in the figures and tables.

17. In suggesting this approach, how do you know the results are sufficient? Are there some measures of quality that can be incorporated into this?

---

## Author Response (AR2)

**EDITOR:**

Editor Decision:

Publish subject to minor revisions (review by editor) (24 Jun 2018) by Stacey Archfield

Comments to the Author:
In reviewing the responses to the reviewers and the revisions in the manuscript, I still find a few areas that need to be addressed before the manuscript can be accepted for publication. I believe these changes are minor but important to ensure the review comments are fully addressed:

1) Both reviewers point out that the abstract is too long. The revisions made by the authors are not sufficient in reducing the length of the manuscript - it appears only a small number of revisions were made aside from making the abstract one paragraph instead of two. The abstract needs to be carefully evaluated so that it only contains the most important points. For example, there is too much detail on the approaches utilized. These sentences can be further shortened (or deleted) and summarized to express only the novel details of the approach.

2) Please spell out uses of "vs." and "e.g".

3) Please add to the new discussion about censored data the following statement (or something like it) in lines 996-997 of the revised manuscript so that the sentence reads: "Thus, overall, the substitution did not substantially affect the interpretation of the considered components; however, we acknowledge that the replacement of censored data with some fraction of the reporting limit is not generally appropriate for dealing with censored data [references provided my the reviewer]." The reviewer had very kindly provided a number of important references on this subject and I do not believe it is adequate to cite only Helsel 2012 and references therein. The authors have done a good amount of work to ensure the effect of the replacement of the censored data did not affect their results and interpretation but I do think the point of the reviewer is important enough to explicitly acknowledge in the text. Not every study does a diligent analysis as the authors did here and it is important that the authors convey to the readers that they are aware of the literature on this topic.

**AUTHORS**:

Dear Stacey,

thank you very much for the positive assessment of the revised version of our manuscript and your comments to further improve the manuscript.

We included your suggestions in the revised manuscript as follows:

1) We further shortened the abstract. Please see the updated abstract based on the version of the 8th of May 2018 in our response to comment #2 of Referee 1.

2) We spelled "vs." out. After a follow-up email with the Editor and evaluation of HESS submission guidelines, no changes were made to the use of "e.g.".

3) We added the statement as you suggested and cited all references the referees suggested, except from Helsel (2005) "Nondetects and Data Analysis" as this is the first edition of Helsel (2012) "Statistics for Censored Environmental Data Using Minitab and R". Please see our updated response to comment #7 of Referee 2.

Please find below our updated final responses to all comments of the referees and the manuscript with the marked changes. Please note that we did not explicitly state the formal changes (spelling, remove of spaces, updating of links) we applied in the revised manuscript in our updated final responses. The different colors in our responses to the referees mark:

- unaltered responses from the 8th of May 2018,
- new or modified responses from the 1st of June 2018,
- changes according to your suggestions from the 24th of June 2018.

Best regards

Christian Lehr

(on behalf of the authors)

**EDITOR:**

Editor Decision:

Publish subject to minor revisions (further review by editor) (24 May 2018) by Stacey Archfield

Comments to the Author:
The manuscript has received two reviews. Both reviewers had a positive reaction to the manuscript and felt that the manuscript was appropriate for publication in HESS, subject to what I would characterize as minor revision based on the excellent and constructive review comments.

The authors have responded to each of the comments and I would now instruct the authors to make their proposed changes to their manuscript. It was not clear from the author responses how Comments #3 and 4 by Reviewer 1 will be addressed in the manuscript so I ask that the authors indicate this in their final responses to the reviewer comments.

I look forward to final acceptance of this manuscript once the review comments have been addressed.

Thank you for considering HESS for your work,
Stacey

**AUTHORS**:

Dear Stacey,

thank you very much for the information about the positive assessment of our manuscript.

We made the proposed changes to the manuscript and added the missing information on the changes in the manuscript regarding comments # 3 and # 4 of referee 1 in our final response to the referees. We hope that those proposed changes likewise covered the raised issues. During revision, we realized one detail concerning the results at site S_121 which might irritate the reader. We included this in our response to comment # 4 b of referee 1 to further clarify how the local water quality relationships at a site or of single samples relate to the overall picture constituted by the components.

Please find our detailed final responses to all comments of the referees and the manuscript with the marked changes. For our final responses we marked the unaltered responses from the 8th of May 2018 with blue font color. New or modified responses are marked with red font color. Please note that we did not explicitly state the formal changes (spelling, remove of spaces, updating of links) we applied in the revised manuscript in our final responses. The series of apparently double figures in the marked up version of the manuscript are caused by the new treatment of missing values according to comment # 7 of referee 1.

We want to thank once again both referees for their positive feedback, their constructive comments and their time.

The comments helped us a lot to improve the quality and clarity of the manuscript and made us think about some aspects of our study in more detail.

Best regards

Christian Lehr

(on behalf of the authors)

**Anonymous Referee #1**

**REFEREE:** This manuscript presents a new exploratory framework for detection of dominant changes in multivariate water quality data sets with irregular sampling in space and time. The paper is well written and I think it is a valuable contribution to the hydrological community. I recommend its publication after the following comments are addressed.

General comments:

1. On the novelty of the proposed framework: I think this manuscript can foster future research ideas and efforts that are aimed toward detecting dominant changes in watershed using multivariate data at multiple sites. I think this type of coherent and systematic investigation of watershed data is limited in the literature, since previous studies have tended to focus on either only a few sites or a few constituents.

**AUTHORS:** We thank the referee very much for these very positive statements!

**REFEREE:** 2. On the abstract: I found it quite lengthy (469 words), which prevents readers from quickly grasping the key messages. Also, it is not customary to have more than one paragraph in the abstract.

**AUTHORS:** We shortened it and reformatted it to one paragraph. The new abstract (339 words) reads:

"Time series of groundwater and stream water quality often exhibit substantial temporal and spatial variability , whereas typical existing monitoring data sets, e.g. from environmental agencies, are usually characterized by relatively low sampling frequency and irregular sampling in space and / or time. This complicates the differentiation between anthropogenic influence and natural variability as well as the detection of changes in water quality which indicate changes of single drivers. We suggest the new term 'dominant changes' for changes in multivariate water quality data which concern 1) multiple variables, 2) multiple sites and 3) long-term patterns and present an exploratory framework for the detection of such dominant changes in  data sets with irregular sampling in space and time. Firstly,  a non-linear dimension reduction technique  was used to summarize the dominant spatiotemporal dynamics in the multivariate water quality data set in a few components. Those were used to derive hypotheses on the dominant drivers influencing water quality. Secondly, different sampling sites were compared with respect to median component values. Thirdly, time series of the components at single sites were analysed for long-term patterns.

 We tested the approach with a joint stream water and groundwater data set quality consisting of 1572 samples, each comprising sixteen variables, sampled with a spatially and temporally irregular sampling scheme at 29 sites  in northeast Germany from 1998 to 2009. The first f components were  interpreted as 1) agriculturally induced enhancement of the natural background level of solute concentration, 2) redox sequence from reducing conditions in deep groundwater to post oxic conditions in shallow groundwater and oxic conditions in stream water, 3) mixing ratio of deep and shallow groundwater to the streamflow and 4) sporadic events of slurry application in the agricultural practice. Dominant changes were observed for the first two components. The changing intensity of the $1^{st}$ component was interpreted as response to the temporal variability of the thickness of the unsaturated zone. A steady increase of the $2^{nd}$ component at most stream water sites pointed towards progressing depletion of the denitrification capacity of the deep aquifer."

**REFEREE:** 3. On the coverage of the monitoring data: The paper addresses the 'time' aspect of the collected water quality data but lacks a thorough discussion on the 'discharge' and 'season' aspects of those data. Were all constituents at these sites sampled roughly similarly across season? Were they sampled roughly similarly during normal-flow and stormflow conditions? Such information is important and can be simply shown with boxplots (e.g., with "month" and "discharge percentiles" as x-axes respectively.) If samples at these sites were not taken roughly similarly across season or discharge, how would that affect the validity of the proposed exploratory framework and the interpretation of the results? The authors should comment on that.

**AUTHORS:** The monitoring did not explicitly distinguish between normal-flow and stormflow conditions. It rather aimed to fulfill the approximately monthly sampling frequency in the streams. Each sample contained all 16 constituents except for the missing values (Table S3). The grab samples were taken on the days marked in the left panel of Figure 2. Thus, while there are definitely irregularities among the series and within series over the course of time, the sites were sampled roughly similarly across season. The most important systematic deviation from this rule were the Peege sites and the most upstream sites of the Quillow, which often desiccate in summer (p. 36, l. 780-782).

In general, the interpretation of the components should consider the temporal structure of the data set. E.g. systematic deviations as the ones describe above should be considered. Thus, we included it in our interpretation of the $1^{st}$ component (p. 36, l. 778 et seq.).

If the monitoring would in general not have been performed roughly similarly across seasons, e.g. if one or more seasons would in general be missing, the estimation of the seasonality would not be applicable. If the monitoring would be such, that there would be different seasons sampled in different years, this would have to be considered in the estimation of the trend.

We agree that considering discharge data would be valuable. Unfortunately, the monitoring did not include discharge measurements. The monitoring aimed to cover the spatial and temporal variability  of water quality along the Quillow stream its tributaries and the adjacent streams. Discharge data was only available at sites Q_93 and S_118. Thus we did not include it in the presented analysis.

To clarify this we changed the last sentence of the third paragraph of section 2.2 in the revised manuscript to:

"In total, sampling intervals between two consecutive samples varied between nine and 714 days (Figure 2). The sites were sampled roughly similarly across seasons (left panel Figure 2). The most important systematic deviation from this rule were the Peege sites and the most upstream sites of the Quillow (left panel Figure 2 and Figure 1), which often fall dry in summer (Merz and Steidl, 2015)."

We changed the third sentence of the fourth paragraph of section 5.3 to:

"This leads often to drought in the uppermost stream reaches (left panel Figure 2; Merz and Steidl, 2015)."

We changed the first sentence in the first paragraph of section 2.2 to:

"The monitoring aimed to cover the spatial variability and temporal of water quality along the Quillow stream, its tributaries and the adjacent streams. The main focus of the monitoring was the Quillow catchment."

We added a new sixth sentence to the first paragraph of section 2.2:

"Discharge data was only available at sites Q_93 and S_118 (Figure 1). Thus we did not include it in the presented analysis."

And we added a new last paragraph in section 5.4:

"In general, the interpretation of the components should consider the temporal structure of the data set. For example in this study the drying out of the streams at the Peege sites and the most upstream sites of the Quillow in summer was the most important systematic deviation from an otherwise roughly similar sampling across seasons (left panel Figure 2). This information was included in the interpretation of the 1st component (section 5.3). If the monitoring would in general not have been performed roughly similarly across seasons, e.g. if one or more seasons would in general be missing, the estimation of the seasonality would not be applicable. If the monitoring would be such that there would be different seasons sampled in different years, this would have to be considered in the estimation of the trend."

**REFEREE:** 4. On the general applicability of the framework: Several points shall be discussed by the authors regarding the applicability of the framework, which can guide its application to monitoring network elsewhere.

a) Is the framework intended to solute data only? Sediment and total phosphorus are typically monitored by many programs. Do the authors recommend the inclusion of such constituents in the proposed framework?

**AUTHORS:** Technically it is possible to include other data than solutes. However, the multivariate components derived by the dimension reduction approach are at the basis of our interpretation. Thus including other types of data might in some cases complicate the interpretation.

In general we would not mix variables with different scales of measures (e.g. nominal variables and ratio scaled variables).

For the interpretation we recommend to keep in mind, that all included variables are z-scaled prior to the dimension reduction. Thus all of them are equally weighted. For example if we would include only one sediment variable to our set of 16 water quality solutes, we expect that it would not change too much of the derived components.

To address this issue, we extended the fourth paragraph in section 5.5 to the following two paragraphs:

"Technically it is possible to combine other data than solutes (e.g. sediment data, biological indicators, etc.) together with the solutes in one joined data set for the derivation of the components. However, the multivariate components derived by the dimension reduction approach are the basis of the subsequent interpretation of the results. It has to be considered as well that all included variables are equally weighted due to the z-scaling prior to the dimension reduction. Thus, including other types of data might in some cases complicate the interpretation. In general, we recommend not to mix variables with different scales of measures (e.g. nominal variables and ratio scaled variables) in the data base for the derivation of the components.

Instead, data which was not used in the derivation of the components can be used as additional information for their interpretation. For example in this study, we used in addition to the spatiotemporal features of the components other variables like groundwater level series, $Fe_2^+$ and $HCO_3^-$ concentration from the groundwater samples, the spatial distribution of land use, and expert knowledge on the study area for the derivation of the hypotheses. A thorough…"

**REFEREE:** b) What is the threshold for a constituent (or a site) to be included in the analysis? Specifically, how many samples are required for a constituent-site pair to be eligible? I am puzzled by the few stations in Figure 2 that have only 1-8 samples. I wonder whether these site-constituent pairs should be disregarded.

**AUTHORS:** It depends on the focus of the study which samples might be considered neglectable. In our case the reasoning was to provide an exploratory approach which enables to get an overview on as much of the available data as possible without too many decisions beforehand which samples / sites to disregard (see also first sentence of comment 4c of referee 1). We intentionally included all samples

available, as long as not more than two of the 16 monitored variables were missing (p. 11, l. 221-223). If the data is organized as in our application, that means that the solutes serve as variables and the samples as observations, than the dimension reduction approach is "blind" to the information which sample belongs to which site. This information is maintained as index of the samples / observations. It is used for example to calculate for each site the median of the component values (section 4.2) and to assess at each site the representation of interpoint distances from the original data space in the low-dimensional projection (Table S4).

Because the selection of data points at a site is only a subset of the global data set for which the dimension reduction was performed, the performances regarding the representation of interpoint distances differ between the individual sites (Table S4) as well as compared to the overall performance for the global data set (Table 2). At some sites it can even happen that adding more components does not for every component improve the representation of interpoint distances in the low-dimensional projection. This occurred in this study only at site S_121 where the representation of interpoint distances with four components ($R^2$ = 0.68) was slightly worse than with three components ($R^2$ = 0.66) (Table S4). This indicated an anomaly at this specific site compared to all other sites with respect to the 4$^{th}$ component, respectively the solutes which mainly determine the 4$^{th}$ component. We traced this phenomenon back to one single sample from the 25$^{th}$ of May 2004 which comprised relatively high DOC values and at the same time relatively low values of $K^+$, which is opposing the relationships indicative for the 4$^{th}$ component (Figure 3). The deterioration of the representation of the interpoint distances after adding the 4$^{th}$ component at this site vanished in an Isomap analysis which was performed without this sample. We were not able to find an explanation for this exceptional sample. However, it underlined that by applying a dimension reduction method every single sample is put into perspective of the global features of the data set as depicted by the components.

This interplay between the local perspective on the water quality relationships in the subsets of the sites or in individual samples on the one hand and of the global perspective of the whole data set on the other hand is a key feature of the presented analysis. The derived components constitute a frame in which all samples are integrated independent of the number of sample per site. Thus, in our application we get the information of how those sites with very little samples group or behave in relation to the others. Even a few samples might indicate e.g. that the respective site behaves similar to other sites with respect to some components and very different with respect to other components. This information would be lost if those samples would be excluded beforehand.

To clarify this, we added the following sentences after the fourth sentence in the second paragraph of section 3.3.2:

"For the local assessment of representation of interpoint distances at the individual sites, only the data points from the respective sites were used. Because the selection of data points at a site is only a subset of the global data set for which the dimension

reduction was performed, the performances regarding the representation of interpoint distances differ between the individual sites as well as compared to the overall performance for the global data set. At some sites it can even happen that adding more components does not for every component improve the representation of interpoint distances in the low-dimensional projection."

We replaced the last two sentences in the last paragraph of section 5.2 with:

"However, the performance of the representation of the interpoint distances after adding the 4[th] component differed substantially between the different sites (Table S4). In case of site S_121 the representation of interpoint distances with four components ($R^2$ = 0.68) was even slightly worse than with three components ($R^2$ = 0.66) (Table S4). This indicated an anomaly at this specific site compared to all other sites with respect to the 4[th] component, respectively the solutes which mainly determine the 4[th] component. We traced this phenomenon back to one single sample from the 25[th] of May 2004 which comprised relatively high DOC values and at the same time relatively low values of $K^+$, which is opposing the relationships indicative for the 4[th] component (Figure 3). The deterioration of the representation of the interpoint distances after adding the 4[th] component at this site vanished in an Isomap analysis which was performed without this sample. We were not able to find an explanation for this exceptional sample. However, it underlined that by applying a dimension reduction method every single sample is put into perspective of the global features of the data set as depicted by the components. Overall, the 4[th] component underlines the necessity to develop and use methods in environmental data analysis which enable to consider non-linear processes as well as singular and site-specific events."

We moved the last paragraph of section 5.4 as new first paragraph. And we rewrote the former first paragraph as new second paragraph continuing as the new beginning of the third paragraph. The  rewritten second and third paragraph reads:

"It is important to note that our approach does not require the same number of samples per site (Figure 2). The derived components constitute a frame in which all samples are integrated independent of the number of sample per site. Thus, in our application we get the information of how those sites with very little samples group or behave in relation to the others. Even a few samples might indicate for example that the respective site behaves similar to other sites with respect to some components and very different with respect to other components. The influence of single samples for the integration of the different sites into the global pattern of the water quality relationships summarized by the 4[th] component is an illustrative example for that (section 5.2). Thus, even occasional sampling at some sites helps assessing the strength of effects of the respective drivers at these sites and might support or contradict hypotheses on spatial variability and related long-term patterns of those influences. This information would be lost if those samples would be excluded beforehand.

In addition, the approach followed here does not require identical temporal sampling resolution at all sites or synchronous sampling dates. Thus, a strictly regular sampling design, which is hardly feasible, is no prerequisite. Correspondingly, data from different monitoring programs could be used for a joint analysis. Sampling intervals …"

**REFEREE:** c) For such multi-site and multi-constituent exploration, all available data should be considered to enhance the robustness of modeling results. However, not all the data are consistently available across the sites. Then, how should one handle the tradeoff between the number of constituents and the number of sites? If we rank all constituents by the number of applicable sites, C1, C2, C3, C3, . . .., C16, then what is the relative gain of sequentially adding extra constituents (from C1 to C16) into the analysis framework? Can an explicit rule be developed to prevent adding new constituents to the framework?

**AUTHORS:** Again, this depends on the focus of the study. In our case we aimed to maintain the spatial coverage of the monitoring. If the main focus is to get an understanding of the multivariate water quality dynamics in detail, it might be worthwhile in the sketched trade-off scenario to disregard some sites and gain some constituents.

We have not thought about an explicit rule to prevent adding new constituents so far. But what we think could be considered is a correlation analysis of all variables beforehand to rule out the variables that correlate stronger than a pre-defined threshold. However, we recommend not to stick only to the threshold, but to visually examine the scatterplots of the respective variables to check for systematic deviations from the global relationship. There might be e.g. some sites or seasons in which the otherwise tight relationship gets weaker.

What we did is to exclude the variables with  more than 5% missing values (p. 10, l. 218-219) to keep the possible effect of any method of replacement rather low.

We included those considerations as two new paragraphs in section 5.5 after the new paragraph related to comment 8 of referee 1 and comment 7 of referee 2 and prior to the new paragraph related to comment 4a of referee 1:

"For data sets in which the number of measured variables differs between the sites there is a trade-off between number of considered variables versus number of considered sites. Depending on the focus of the study different selections of the data set can be used. For example if the main focus of the study is to analyse the multivariate water quality dynamics in detail it might be worthwhile to disregard some sites to be able to include more variables. If the focus is to maintain the spatial coverage of the monitoring, like in this study, more sites might be of more value than additional variables. Depending on the available resources a third option would be to perform two analyses, one focusing on more variables, one on more sites, and comparing the results. If it is possible to link the considered components, like we did in the preceding paragraph, this proceeding can be used for spatial extrapolation of

the hypotheses derived from the version which included more variables. However, in our case the sketched trade-off was not dramatic. Thus, we excluded only the variables with more than 5% missing values (section 2.2) to keep the possible effect of any method of replacement rather low.

To prevent adding variables with little information gain it is recommendable to perform a correlation analysis beforehand and rule out highly correlated variables. For this purpose we recommend not to rely only on a numerical measure of correlation, but to visually examine the scatterplots of the respective variables to check for systematic deviations from the global relationship. There might be e.g. some sites or seasons in which the otherwise tight relationship gets weaker pointing to local or temporal phenomena."

**REFEREE:** 5. On the irregularity nature of the monitoring data: The authors have provided adequate references in many parts of the manuscript. One exception is on the irregularity of water quality data (~ line 110 and also Section 5.4). One reference that you may find useful is provided below, which discusses at least two points that are discussed in this manuscript, including (a) irregularity nature of water quality data and how to model that property and (b) fractal scaling in water quality data which may affect trend significance (including the trend approaches used here).

Zhang, Q., Harman, C. J., and Kirchner, J. W. (2018), Evaluation of statistical methods for quantifying fractal scaling in water-quality time series with irregular sampling, Hydrol. Earth Syst. Sci., 22, 1175-1192, https://doi.org/10.5194/hess-22-1175-2018.

**AUTHORS:** We included the suggested reference in the revised manuscript at the end of the fourth paragraph in the introduction:

"Thus, in environmental monitoring practice, data sets with gaps and periods with corrupted measurements are more the rule rather than the exception (c.f., e.g., Zhang et al., 2018 for river quality data)."

and in section 3.4.2 as new eigth sentence after the new seventh sentence related to comment # 11 of referee 1:

"Consequently, we did not consider the possible effects of the irregular sampling on the long-term memory (fractal scaling) of the water quality series either (Zhang et al., 2018)."

**REFEREE:** Specific comments:

6. On Figure 2:

a) This is a well-designed figure.

**AUTHORS:** Thank you very much!

**REFEREE:** b) Consider adding vertical reference lines in the right panel to indicate 1-day, 1-week, and 1-month intervals.

c) Add additional reference lines to separate groundwater from stream water – refer to your treatment in Figure 5.

d) Consider using color to distinguish between median and mean.

**AUTHORS:** We updated the figure according to your suggestions.

[Figure]

Figure 2 Left panel: Sampling dates at the sites for the whole monitoring period. Right panel: Boxplots of the variability of sampling intervals during the monitoring period. For better readability, the maximum of the x-axis is limited to 180 days. Median (red) and mean (blue) of sampling intervals are shown separately for the groundwater and stream water sites. Grey vertical lines mark the 1-day, 1-week and 1-month interval. Both panels: The dashed horizontal line separates groundwater sites (bottom) from stream water sites (top). Subscripts: P = Peege, Q = Quillow, S = Strom, St = Stierngraben, U = Ucker, D = Dauergraben, Gs = shallow groundwater, Gd = deep groundwater. The number of samples at each site is given in brackets. Names of the sites with more than 50 samples are printed bold.

**REFEREE:** e) Comment in the text on the apparent outlier in the site GdQ_198 distribution.

**AUTHORS:** This was an exceptional sample taken during maintenance work. We included this information as fourth sentence in the third paragraph of section 2.2 in the revised manuscript:

"The one shorter sampling interval at site GdQ_198 was an exceptional sample taken during maintenance work."

**REFEREE:** f) Do the numbers in bracket represent the number of samples for one constituent or all constituents? Clarify.

**AUTHORS:** The numbers in bracket represent the numbers of samples. Each sample contained all constituents, except for the missing values (Table S3).

We added "Each sample contained measurements of all 16 variables." prior to the sentence "Those water samples…" on page 11 line 221 in the revised manuscript.

**REFEREE:** g) Two of the sites have only one sample each. Justify why those sites should not be removed. In my opinion, those sites which only several samples should also be excluded unless their use can be justified.

**AUTHORS:** We aimed to demonstrate how the suggested exploratory approach can be used irrespective of those rather extreme differences between the numbers of samples per site to get an overview on as much of the available data as possible. While only of indicative value, it still can be interesting to see whether those single sample-sites plot / group different for the different components with respect to the other sites. Please see also our response to comment 4b) of referee 1.

**REFEREE:** 7. Line 248: I would suggest using median for the missing value replacement.

**AUTHORS:** In our case only a small percentage of samples were concerned (in the data set that was used for the dimension reduction at most for DOC: 3.44% and in the only for the comparison used groundwater samples at most $HCO_3^-$: 6.43% Table S3). We compared the two versions (missing value replacement with mean vs. missing value replacement with median). For the PCA, the scores of the first 10 components of the two versions yielded a $R^2 > 0.99$. For Isomap, the first 9 components yielded a $R^2 > 0.99$ and the 10th component a $R^2$ of 0.98. There were only minor differences in the site-specific cumulated $R^2$ of the reproduction of the interpoint distances of the data in the projection by the first four components of Isomap at sites with n > 15 (Table S4). Thus, for our case it did not really make a difference.

However, for other data sets this might be different. Thus, we agree that using median for the missing value replacement is in general the more robust approach.

Therefore, we updated the figures and results in the revised manuscript with the missing values replaced by the median and changed "mean" to "median" in the first sentence of section 3.1.

**REFEREE:** 8. Line 252: Provide references to justify the use of half detection limit for censored values. It is a typical practice but it has been pointed out that such treatment may cause issues to analysis – refer to the references below. This could be a problem for NO2 and PO4, since the two species have significant proportions of censored values (Table S3).

Helsel, D.R., 2006. Fabricating data: how substituting values for nondetects can ruin results, and what can be done about it. Chemosphere, 65(11), pp.2434-2439.

Helsel, D. R. (2005). More than obvious: better methods for interpreting nondetect data. https://pubs.acs.org/doi/pdf/10.1021/es053368a.

**AUTHORS:** Thank you for this substantial comment and the provided references. As both referees raised this point, we will give a joint answer. Please see our response to comment 7 of referee 2.

**REFEREE:** 9. Line 262: How was the threshold of '50 samples' chosen? It is still a small size.

**AUTHORS:** This threshold was a compromise between preferably long time series and the attempt to include preferably many of the series and sites in the analysis, to get an overview on the differences between the sites and catchments. The longest series in our data set comprised 127 samples. Thus, the data set as such is limited in this regard.

**REFEREE:** 10. Line 386 (Eq. 2): Check whether you want to use two equal signs in this equation.

**AUTHORS:** We rewrote the equation. Please see our response to comment 8 of referee 2.

**REFEREE:** 11. Line 421: The effect of autocorrelation on trend analysis is not only relevant to short-memory processes (e.g., AR(1) in Yue et al., 2002), but also long-memory processes (e.g., ARFIMA).

Cohn, T. A., and H. F. Lins (2005), Nature's style: Naturally trendy, Geophys. Res. Lett., 32, L23402, doi:10.1029/2005GL024476.

Zhang, Q., Harman, C. J., and Kirchner, J. W. (2018), Evaluation of statistical methods for quantifying fractal scaling in water-quality time series with irregular sampling, Hydrol. Earth Syst. Sci., 22, 1175-1192, https://doi.org/10.5194/hess-22-1175-2018.

**AUTHORS:** We specified the addressed autocorrelation in the fifth sentence of section 3.4.2 as "short-term autocorrelation" and included the suggested references in the revised manuscript as new seventh sentence in section 3.4.2:

"Neither did we consider long-term memory and its effects on the statistical significance of the trends (Cohn and Lins 2005; Zhang et al., 2018)."

**REFEREE:** 12. Line 456: I think it should be 42% (per Table 2).

**AUTHORS:** 42% is correct. We corrected that in the revised manuscript.

**REFEREE:** 13. Line 459: In addition to temperature, PO4 is also negatively correlated with PC 1.

**AUTHORS:** We decided to mention in the text for each component only the constituents which correlated strongest, because the interpretation was focused on those. The correlation with $PO_4^{3-}$ is negative, but almost zero. That is why we did not mention it. In the same manner, we proceeded for the other components.

**REFEREE:** 14. Line 463: This should be 18% (per Table 2).

**AUTHORS:** 18% is correct. We corrected that in the revised manuscript.

**REFEREE:** 15. Line 537: Check the label for n < 3 in Figure 5, which should not be identical to n < 13.

**AUTHORS:** We changed the label for n < 3 to "X" and reformatted the 4 plots in one column instead of a 2x2 matrix to enable larger labels for better readability.

[Figure]

Figure 5 Boxplots of scores of component 1 to 4 at different sites. Sites with $N_n$ < 13 are marked with '~', those with $N_n$ < 3 with 'X'. Subscripts: P = Peege, Q = Quillow, S = Strom, St = Stierngraben, U = Ucker, D = Dauergraben, Gs = shallow groundwater, Gd = deep groundwater.

**REFEREE:** 16. Line 675: This conclusion should be supported by some references.

**AUTHORS:** We included references and changed the last sentence of the last paragraph of section 5.1 to:

"The catchments of the analysed  streams are only sparsely populated and mainly characterized by intensive agriculture (Table 1). In agricultural landscapes slurry is a typical source in which those nutrients occur in high concentration (Hooda et al., 2000). We are not aware of any other high-concentration sources of this combination of nutrients in the region. The little number of scores with very low scores implied that there were merely single events occurring at some of the sites only. This fits to the finding that the timing of slurry application is crucial for the amount of nutrient loss to the streams (Hooda et al., 2000; Cherobim et al., 2017). Thus, we interpreted the negative peaks of the 4[th] component as sporadic events of slurry application, being either unintentionally directly applied to the stream during the spreading of the slurry or being leached via surface runoff and tile drain discharge after application."

**Anonymous Referee #2**

**REFEREE:** The manuscript proposes an exploratory framework for detection of dominant changes in multivariate water-quality data sets with irregular sampling in space and time. As stated in the introduction, many analysis methods assume regular temporal spacing, but many monitoring networks evolve over time resulting in irregularly spaced samples. The concept is good, some more effort needs to be put into the writing and analysis.

**AUTHORS:** Thank you for the positive statement!

**REFEREE:** 1. The abstract is rather lengthy.

**AUTHORS:** We shortened it. Please see our response to comment 2 of referee 1

**REFEREE:** 2. The introduction contains vague statements and extraneous adverbs. The first sentence of the article is "Numerous high frequency studies unravelled the high temporal variability of stream water quality." This is well known, as shown by the many references. It seems like the first sentence of the article should start with a stronger sentence about the problem at hand.

**AUTHORS:** We added the following sentence as first sentence of the introduction in the revised manuscript:

"Detecting of changes in water quality and the responsible drivers are of fundamental interest for water management purposes as well as for scientific analyses."

**REFEREE:** The second paragraph of the introduction has the phrase "numerous different drivers at different scales." This is vague. Give an example, or qualify the drivers, such as climatic and land-use drivers.

**AUTHORS:** We rewrote the sentence to:

"Instead, a variety of biogeochemical processes (e.g., Stumm and Morgan, 1996; Neal, 2004; Beudert et al., 2015), climatic (e.g., Neal, 2004) and hydrological (e.g., Molenat et al., 2008) variability and anthropogenic influences, for example agricultural (e.g., Basu et al., 2010; Basu et al., 2011; Aubert et al., 2013) or forestal (e.g., Neal, 2004) land use, land use change (e.g., Scanlon et al., 2007; Raymond et al., 2008) or urbanization (e.g., Kroeze et al., 2013), interact at different scales impeding identification of clear cause-effect relationships."

**REFEREE:** The second sentence of the third paragraph is either missing something or "determining" should be "determine."

**AUTHORS:** We rewrote the sentence to:

"Usually only a few dominant processes determine the main dynamics of stream flow, groundwater head or water quality (Grayson and Blöschl, 2000; Sivakumar, 2004; Lischeid et al., 2016)."

and rewrote the sentence on p. 41 l. 929-931 to:

"This is consistent with the prior assumption that there are a few dominant drivers which determine the main stream water and groundwater quality dynamics in the region."

**REFEREE:** 3. In the description of the study area the mean annual precipitation and mean annual temperature are given for the federal state Brandenburg for 1961–1990. This does not overlap with the study period of 1990–2009 at all. With the common use and availability of climatic data, it would not take much effort to report precipitation and temperature for the study period. It is not clear what period the water balance variability values represent.

**AUTHORS:** We replaced the addressed lines in the revised manuscript with:

"At the ZALF weather station Dedelow, which is situated approximately 500m northeast of Q_97 (Figure 1), a mean annual precipitation of 550 mm and a mean annual temperature of 8.9° C was observed for the hydrological years within the study period (1997-11 to 2009-10). The mean annual climatic water balance for this period, calculated from daily precipitation and potential evapotranspiration, was found to be -103 mm, exhibiting high interannual variability with -148 mm in the summer half year and +45 mm in the winter half year."

**REFEREE:** 4. The topography and soils sections are well written and informative.

**AUTHORS:** Thank you very much for this positive feedback!

**REFEREE:** 5. We know the data are collected irregularly, but are they collected to be representative of seasons and flow conditions, i.e., are there high-flow samples?

**AUTHORS:** Thank you for this comment. Referee 1 raised this point as well. Please see our response to comment 3 of referee 1.

**REFEREE:** 6. Figure 2 shows some sites with very little data, yet it seems like they were included. It is not clear how these help inform the method. It seems like there should be some minimum number of samples per year most of the years from 1998 - 2009 in order for a site to be included in the study. Some parts of the proposed framework were done for sites with more than 50 observations. It seems like the entire analysis should be done only with those sites. It is not clear how these low-sample sites fit with the rest of the sites.

**AUTHORS:** Thank you for this comment. Referee 1 raised this point as well. Please see our response to the comments 4.b) and 6.g) of referee 1.

**REFEREE:** 7. It has been very well documented that substituting a fraction of the reporting limit is an inappropriate method for dealing with censored data. See:

Gilliom, R.J., and Helsel, D.R., 1986, Estimation of distributional parameters for censored trace level water quality data, 1. Estimation techniques: Water Resources Reserach, 22, 135–146.

Singh, A., and Nocerino, J., 2002, Robust estimation of mean and variance using environmental data sets with below detection limit observations: Chemometrics and Intelligent Laboratory Systems, 60, 69–86.

Helsel, D. R., 2005, More than obvious - Better method better methods for interpreting nondetect data: Environ. Sci. Technol., 39(20), 419A–423A, DOI: 10.1021/es053368a

Helsel, D.R., 2005, Nondetects and Data Analysis: Wiley-Interscience, 250 p.

Helsel, D.R., 2006, Fabricating data - How substituting values for nondetects can ruin results, and what can be done about it: Chemosphere, 65(11), 2434–2439.

Helsel, D.R., 2012, Statistics for Censored Environmental Data Using Minitab and R: John Wiley & Sons, 324 p.

Admittedly, the percent of censored values is small, but substitution should really not be used anymore in water-quality analyses. I'm not sure if Isometric Feature Mapping can utilize censored values. However, the authors could estimate the mean and standard deviation of the constituents with censored values using regression on order statistics or maximum likelihood methods (see Helsel, 2012) before standardizing the variables. The Akritas-Thiel-Sen median line can be used for the trend analysis.

**AUTHORS:** Thank you for this substantial comment and the provided references. As both referees raised this point, we will give a joint answer.

First of all, we agree that the question of how to deal with censored values is crucial and has to be handled with care.

The censored values in our study are the values below the detection limit of the respective variable, thus the measurements which are considered to be too imprecise to be reported as a single number (according to Helsel, 2012). Still they yield important information, in particular the ratio of values below the detection limit in comparison to values above the detection limit (cf. Helsel, 2012, page 12). This information is provided for all variables in table S3. We agree that censored values are not a big issue for our data set, except for the variables $NO_2^-$ and $PO_4^{3-}$ (and $Fe^{2+}$ for the additional groundwater data, which was not used to calculate the components).

In our case, the purpose of the replacement of values below the detection limit is not to estimate distributional parameters such as mean or standard deviation or to perform statistical tests (like in most applications of the provided references). The

purpose is merely to provide values for all 16 variables in a sample so that the dimension reduction method can be applied.

The standardizing of the variables before applying the dimension reduction method is to achieve equal weighting of the variables. Therefore, the estimation of mean and standard deviation for this purpose has to be based on all values of a variable – whatever values are used for replacement of the censored values.

We are not aware of an isometric feature mapping variant, which is able to explicitly deal with censored values.

Helsel (2012) suggests to perform dimension reduction methods on the rank scaled variables or on a rank based distance matrix if censored values occur. To our understanding, we have to deal here with the trade-off between derivation of more "correct" components (the rank based case) and the loss of information that occurs, in case the ratio scaled variables are transformed to ranks (namely the information on the relative distances of the data points to each other, for example how distant the value of rank x is to the value of rank x-1 in comparison to rank x-2, etc.). For the exploratory purpose of our study, we prefer to maintain this information in the light of the fact that only 2 out of 16 variables are substantially affected.

Although in our case the calculation of the components included the substituted values, the components themselves do not contain censored values any more. Thus, the subsequent time series analysis of the component scores does not have to be designed especially for the treatment of censored values (e.g. Akritas-Thiel-Sen median).

Concerning the correlation of variables and components, we used the residual plots and the spearman rank correlation of residuals and components (Section 3.3.3, p. 17, l. 377-388). We admit that a problem arises with the calculation of the multiple linear regression and therefore the residuals are affected as well. Again, we have to deal with a trade-off between potential information loss regarding the 14 out of 16 variables compared to the more correct treatment of 2 out of 16 variables. Spearman rank is one of the methods recommended by Helsel (2012) for the calculation of correlations with variables with only one reporting limit (in our case the detection limit). However - as in the case of the components - the residuals themselves are calculated with the censored values, but they do not contain censored values as such. For example for $NO_2^-$, the values that were substituted with half of the detection limit would get all the same rank, while the residuals of the linear model of $NO_2^-$ with three of the components do not contain same-ranked values any longer.

Those two decisions (rank-based dimension reduction method yes / no and use of multiple linear regression and the residuals yes / no) can be questioned. Here, we provided arguments, why we did so. Following our argumentation and proceeding, the subsequent time series analysis of component scores as well as the correlation analysis between residuals and components should be not problematic.

In addition, of the two affected variables only $PO_4^{3-}$ is substantial for the interpretation of a component, namely component 4. In this specific case, the range of values of the $4^{th}$ component "was spanned mainly by single large values of $NH_4^+$, $PO_4^{3-}$ and $K^+$ that cannot be explained with the preceding three components (Figure S4). This highlights the importance of particular events for the 4th component." (p.21, l. 483-486). This fits to the distribution of $PO_4^{3-}$ values which exhibits a substantial part of values below the detection limit and some outstandingly large values.

We checked for the influence of the substitution of the two affected variables on the components by performing another PCA and Isomap based on a data set in which $NO_2^-$ and $PO_4^{3-}$ were excluded.

The correlation of the PCA scores of the interpreted components 1 to 4 of version 1 (with $NO_2^-$ and $PO_4^{3-}$) vs. version 2 (without $NO_2^-$ and $PO_4^{3-}$) yielded a $R^2$ of cp1: 0.99, cp2: 0.99, cp3: 0.99, cp4: 0.71.

The correlation of the Isomap scores of the interpreted components 1 to 4 of version 1 (with $NO_2^-$ and $PO_4^{3-}$) vs. version 2 (without $NO_2^-$ and $PO_4^{3-}$) yielded a $R^2$ of cp1: 0.99, cp2: 0.98, cp3: 0.97, cp4: 0.64.

The same correlations were found for a third version in which $NO_2^-$ and $PO_4^{3-}$ were excluded and all missing values were replaced with the respective median, instead of the mean as suggested by Referee 1 in Comment 7.

The comparison of the two versions with respect to the Spearman rank correlations of Isomap scores of the first four components and the residuals (please see Figure 3 in the manuscript for the respective values of version 1) yielded a $R^2$ of cp1: 0.98, cp2: 0.99, cp3: 0.99, cp4: 0.88.

Thus the first three components are virtually identical. The fourth component is affected, because $PO_4^{3-}$ is one of the important variables determining this component. Still, the similarity of the correlations of Isomap scores and component 4 of both versions suggest that even for this component the variables $NO_2^-$ and $PO_4^{3-}$, and therefore the substitution of values below the detection limit with half of the detection limit, did not substantially affect the derived components.

To summarize:

We agree that the treatment of censored values is an issue that has to be considered carefully, in our case especially for $NO_2^-$ and $PO_4^{3-}$. We decided for our data set and the amount of affected values / variables to go not for a rank based dimension reduction method, due to the loss of information. Therefore, we needed to provide numerical values for the values below the detection limit. We decided to choose half the detection limit as a simple marker. The calculation of the components, the multiple linear regression and the residuals is affected by the substitution. We showed that for our case the substitution did not substantially affect the interpretation of the results.

We included the following paragraph after the third paragraph in section 5.5. "Exploratory framework" in the revised manuscript:

[revised manuscript text omitted]

**REFEREE:** 9. In the interpretation of components, the authors describe using multiple linear regression, which is a parametric method that assumes a model linear in the parameters, but then make an argument for a non-parametric measure of correlation applied to the multiple linear regression results. This seems contradictory.

**AUTHORS:** The residuals of the multiple linear regression were used to exclude the influence of the respective other three components in the assessment of correlation between single variables and components (p. 17, l. 377-387). Thus, the aim was to facilitate the assessment of the specific contribution of a single component out of the four considered components, especially in the visual examination of the residual-plots (p.17, l. 387).

To summarize the relationships between residuals and components we used Spearman rank correlation (p.17, l. 388+389). Most of the global relationships in this study were linear (Figure S1-S4). This is usually not known beforehand. Using Spearman rank correlation enabled to consider non-linear relationships between residuals and components as well, as long as they are monotonic.

However, the main benefit in this study was that Spearman rank correlation is less sensitive to extreme values compared to Pearson correlation. This concerned especially the assessment of the relationships of the residuals of $SO_4^{2-}$ and $Cl^-$ with the 2nd component and the residuals of $PO_4^{3-}$ and $NH_4^+$ with the 4th component

(Figure S2 and S4), which were way stronger expressed with Pearson correlation due to a few single extreme values.

In addition, if the step with the multiple linear regression is omitted, thus if the correlations between variables and components are assessed based on the measured variables and not the residuals, than the use of Spearman rank correlation yields the additional benefit that it can deal with censored values (because there is in our case only one detection limit per variable ➜ cf. Helsel, 2012, p. 218).

To clarify this issue we will replace the last sentence in section 3.3.3 with:

"To summarize the relationships between components and residuals we used Spearman rank correlation, which enables to consider non-linear relationships as well, as long as they are monotonic. Besides, it is less sensitive to extreme values than Pearson correlation."

and the 2$^{nd}$ sentence in the 3$^{rd}$ paragraph of section 5.5 with:

"Again, whether the relationships are linear, as it was for most of the global relationships in this study (Figure S1-S4), is usually not known beforehand. Summarizing the relationships between residuals and components with Spearman rank correlation enables to consider non-linear relationships between residuals and components as well, as long as they are monotonic. However, the main benefit in this study was that Spearman rank correlation is less sensitive to extreme values compared to Pearson correlation. This concerned especially the assessment of the relationships of the residuals of $SO_4^{2-}$ and $Cl^-$ with the 2$^{nd}$ component and the residuals of $PO_4^{3-}$ and $NH_4^+$ with the 4$^{th}$ component (Figure S2 and S4), which were way stronger expressed with Pearson correlation due to a few single extreme values."

**REFEREE:** 10. Consider presenting the methods and the results in the same order for parallel construction.

**AUTHORS:** Thanks for this comment. We tried different ways to structure the manuscript during the writing process before we ended up with the current structure. The reasoning was to firstly introduce separately all the tools in the methods section before we secondly present the results from the perspective of the different aspects of the dominant changes in the data set.

We still think that it is a reasonably compromise for the purpose of this study. The structures of the methods and results sections are not parallel as you mentioned. Instead, we explicitly introduced the structure of the results and discussion section in the section "3.2 Exploratory framework". The purpose of this section is to wrap up all the methods in one consistent picture and illustrate the workflow.

**REFEREE:** 11. In the discussion, the conclusions on page 32 about the 1st component were not well supported. There were a lot of statements like "we assume a general effect," some process "might" happen, some processes "tend to enhance."

The discussion of the 2nd component was better supported with information about the sediments in the area. Some of the material in the first paragraph of section 5.2 should be moved up to better support the conclusions about the 1st component.

**AUTHORS:** Interpretations of the components were developed in a systematic way, considering the aspects of the correlations of variables and components (section 5.1), the spatial patterns (section 5.2) and the temporal patterns (section 5.3) of the components. Any interpretation is not only based on section 5.1 but after putting the different pieces of information in section 5.1, 5.2 and 5.3 together (p. 17, l. 374-376). We would like to stick to this structure for the sake of clarity. As guidance for the reader, we present the hypotheses for the components already in section 5.1. Correspondingly, we formulated the hypothesis for the 1$^{st}$ component in section 5.1 in a careful manner, to express that the aspect of correlation among the solutes alone is merely one aspect which needs further support. This is realized in sections 5.2 and 5.3 in which we add the spatial and temporal patterns to the picture to strengthen our hypothesis.

To more explicitly state the background of our hypothesis for the 1$^{st}$ component in this early stage of the argumentation, we added a new introductory sentence for the 3$^{rd}$ paragraph in section 5.1 in the revised manuscript.

"The whole study region is characterized by relatively intense agriculture (Table 1)."

**REFEREE:** The discussion of the 4th component on page 33 seemed speculative. Has this been modelled or shown elsewhere?

**AUTHORS:** Thank you for this comment. Referee 1 raised this point as well. Please see our response to the comment 16 of referee 1.

**REFEREE:** 12. Page 37 states nicely some important implications of the observed water quality.

**AUTHORS:** Thank you very much for this positive feedback!

**REFEREE:** 13. Page 40, line 895, change "is" to "are."

**AUTHORS:** The "is" refers to "The assessment of ….. is less sensitive…"

**REFEREE:** 14. Page 40, line 901, "Complementary" does not seem like an appropriate word for this sentence.

**AUTHORS:**

OLD RESPONSE from 8 May 2018:

NEW RESPONSE:

Considering the comment 4a) of Referee 1 we rewrote the addressed sentence. Please see our response there.

**REFEREE:** 15. Some of the results, discussion, and conclusions mention both PCA and Isomap, but some of the numbers, figures, results must come from one of them specifically. That should be made more clear.

**AUTHORS:** PCA is used here merely as a benchmark for the Isomap results (p. 15, l. 316+317) and to introduce the concept / functioning of dimension reduction methods to the reader, as we expected it to be more familiar to the hydrological community. To our knowledge it is the most established and most used dimension reduction method in hydrology. Another reason why we included it in the study is because some readers might want to apply the framework based on PCA alone.

Thus, all presented and discussed results are from Isomap except from the "benchmark" comparison with PCA (Table 2).

We clarified this in the revised manuscript. We moved the last sentence of the first paragraph of section 5.5 to the beginning of 5.1 and added another sentence:

"Non-linear Isomap performed in this study only slightly better with respect to the representation of interpoint distances than PCA (Table 2), suggesting that mainly linear relationships were of importance for the overall dynamics in the data set. As there were only minor differences, we will present in the following the results of Isomap only."

The  first sentence in the second paragraph of section 5.5 reads now:

"Whether the relationships in the data set are mainly linear ones, as in this study, or whether there are considerably non-linear relationships as well, is usually not known in advance."

**REFEREE:** 16. Check that numbers in the text agree with the numbers in the figures and tables.

**AUTHORS:** We carefully checked the manuscript. Unfortunately we missed the two numbers referee 1 pointed out (comment 12 and 14 of referee 1).

**REFEREE:** 17. In suggesting this approach, how do you know the results are sufficient?

**AUTHORS:** In our understanding, the sufficiency of the results depends on the purpose of the study.

Our purpose was to provide a framework for the exploratory analysis of dominant changes in the spatial and temporal features of multivariate water quality data sets. We think that we were able to demonstrate its applicability with the presented study.

**REFEREE:** Are there some measures of quality that can be incorporated into this?

A very basic measure of quality is to measure the amount of variance in the data set, which is assigned to the first components. For example a more or less evenly distributed variance among the first components indicates that there are no dominant structures in the data set the used method is sensitive for. This result in itself can be rather interesting. Apart from that it would be in this case most probably not possible to link the components to drivers which help to better understand the monitored system.

A next step can be to compare the results of different dimension reduction methods, as we did here with principal component analysis and isometric feature mapping (Table 2). If applicable, the results of the dimension reduction method can be evaluated with different performance measures (e.g. the PCA performance can be evaluated with the "classical" approach via the sizes of the eigenvalues that are assigned to the components, or the correlation of the distance matrices of the analysed data in the original data space and the projection, as it was done in this study).

Concerning the interpretation of the components, we want to emphasize once more that the suggested approach is an exploratory one. Testing the derived hypothesis - for example by correlating the results with additional data - is a next step. Another option would be to test the hypotheses with virtual or "real-life" experiments (p. 40, l. 907-909).

Depending on the structure of the data set (e.g. its spatial and temporal resolution, number of samples per site, etc.) one option could be to perform the suggested approach with different subsets of the data set and compare the derived spatial and temporal patterns for example for different regions or time periods. The same approach can be used to check the results for their dependence on specific selections of the data set, which can serve as an estimation of the representativeness of the results for the overall region and time period.

[revised manuscript text omitted]